# Metabolic and non-metabolic liver zonation is established non-synchronously and requires sinusoidal Wnts

**Ruihua Ma, Angelica S Martínez-Ramírez, Thomas L Borders, Fanding Gao, Beatriz Sosa-Pineda***

Feinberg Cardiovascular and Renal Research Institute, Department of Medicine, Northwestern University Feinberg School of Medicine, Chicago, United States

**Abstract** The distribution of complementary metabolic functions in hepatocytes along a portocentral axis is called liver zonation. Endothelial secreted Wnt ligands maintain metabolic zonation in the adult murine liver but whether those ligands are necessary to initiate zonation in the immature liver has been only partially explored. Also, numerous non-metabolic proteins display zonated expression in the adult liver but it is not entirely clear if their localization requires endothelial Wnts. Here we used a novel transgenic mouse model to compare the spatial distribution of zonated non-metabolic proteins with that of typical zonated metabolic enzymes during liver maturation and after acute injury induced by carbon tetrachloride ($CCl_4$). We also investigated how preventing Wnt ligand secretion from endothelial cells affects zonation patterns under homeostasis and after acute injury. Our study demonstrates that metabolic and non-metabolic zonation are established non-synchronously during maturation and regeneration and require multiple endothelial Wnt sources.

*For correspondence:
beatriz.sosa-pineda@
northwestern.edu

**Competing interests:** The authors declare that no competing interests exist.

## Introduction

The liver executes multiple functions involved with anabolic and catabolic metabolism, bile and protein secretion, drug detoxification, glycogen storage, and other processes. These tasks are performed simultaneously and with maximal energy-efficiency thanks to a unique anatomical architecture consisting of hexagonal functional units (a.k.a. 'lobules'), typically arranged in a honeycomb-like pattern (*Gebhardt and Matz-Soja, 2014*). At the periphery of the lobules, blood enters the portal vein and the hepatic artery and flows into the liver sinusoids towards the central vein. Inside the lobules, the highly polarized hepatocytes assemble into cords that face the hepatic sinusoids on the basolateral side and form canaliculi on the apical side (*Tanaka and Miyajima, 2016*). Although morphologically quite similar, hepatocytes accomplish very different metabolic roles according to their specific position along the lobular portocentral axis. The segregation of hepatocytes into discrete functional areas is referred to as 'Liver Zonation' and is the basis for the partition of the hepatic lobule into three metabolic zones: periportal Zone 1 consisting of 6–8 hepatocyte layers that receive blood enriched in oxygen and nutrients and control glycogen metabolism, amino acid utilization and ammonia detoxification; intermediate perivenous Zone 2 consisting of 6–10 hepatocyte layers with a major role in xenobiotic metabolism; and perivenous Zone 3 formed by 2–3 hepatocyte layers that surround the central veins and perform biotransformation reactions, glutamine synthesis and glycolysis (*Jungermann and Katz, 1989*). Metabolic zonation optimizes liver physiology by quickly adapting this organ to exogenous nutritional challenges and endogenous metabolic demands. Moreover, pathologic alterations that permanently change zonation patterns lead to metabolic disease (*Gebhardt and Matz-Soja, 2014*).

Pioneer studies revealed that canonical Wnt/β-catenin signaling preserves metabolic zonation in the adult liver (*Benhamouche et al., 2006*; *Burke et al., 2009*; *Planas-Paz et al., 2016*; *Sekine et al., 2006*; *Yang et al., 2014*) and promotes the expression of pericentral enzymes involved with glutamine synthesis in the late fetal liver (*Burke et al., 2018*). More recent investigations also demonstrated that Wnt ligands and the Wnt agonist R-spondin3 produced by hepatic endothelial cells are necessary to preserve metabolic zonation in the adult liver (*Leibing et al., 2018*; *Planas-Paz et al., 2016*; *Preziosi et al., 2018*; *Rocha et al., 2015*; *Wang et al., 2015*). While these and other studies have increased our knowledge of adult metabolic zonation, we still do not know much about how metabolic and non-metabolic proteins begin to segregate into specific regions in the maturing liver or how zonated protein expression is reestablished in hepatocytes following acute hepatic injury. Moreover, it has been shown that hepatic sinusoidal cells produce Wnt ligands (*Ding et al., 2010*; *Wang et al., 2015*) but the potential contribution of this specialized endothelium to liver zonation recovery after acute injury has not been directly proven.

Here we used a novel transgenic mouse strain (*Cldn2-EGFP*) (*Gong et al., 2003*) to investigate how the tight junction protein claudin-2 becomes zonated in hepatocytes during liver maturation and after $CCl_4$-induced acute injury. We also used conditional deletion approaches and *Cldn2-EGFP* mice to investigate the role of hepatic sinusoid-derived Wnt ligands in the establishment and maintenance of liver zonation, and the role of endothelial-derived Wnt ligands in zonation restoration after $CCl_4$-induced damage.

## Results

### Spatiotemporal analysis of postnatal liver zonation using *Cldn2-EGFP* transgenic mice

Published studies showed that the tight junction protein claudin-2 exhibits zonated expression in perivenous areas of the adult murine liver (*Matsumoto et al., 2014*; *Rahner et al., 2001*). Using immunohistochemistry analysis, we corroborated claudin-2 expression almost exclusively in perivenous areas in the adult mouse liver (*Figure 1A*). In addition, we discovered that this protein is expressed throughout this organ of mice at postnatal [P] day 2 (*Figure 1A*). To further examine how claudin-2 hepatic expression transitions from birth to adulthood, we took advantage of *Cldn2-EGFP* mice that express a *green fluorescence protein* (*GFP*) transgene under the control of *Cldn2* regulatory elements (*Gong et al., 2003*). Our immunostaining results showed identical distribution of claudin-2 and GFP in various tissues of wildtype and *Cldn2-EGFP* mice dissected at P2 or 6 months of age (including the liver [*Figure 1A*], intrahepatic bile ducts [*Figure 1—figure supplement 1A*], extrahepatic biliary tissues [*Figure 1—figure supplement 1B*] and gut [data not shown]). Also, double-immunofluorescence staining of 1–6 months old *Cldn2-EGFP* livers showed co-expression of GFP with various hepatocyte markers (i.e., HNF4α, Tbx3 and Prox1; *Figure 1B*), the Zone 3 marker GS (*Figure 1B*) and the Zone 2/3 marker Cyp2e1 (*Figure 1B*). In contrast, GFP expression was very low or negligible in periportal hepatocytes expressing the cell adhesion protein and Zone 1 marker E-cadherin (*Figure 1B*). These results demonstrate that GFP expression faithfully recapitulates the endogenous expression of claudin-2 in many tissues of *Cldn2-EGFP* mice.

Next, we compared the spatiotemporal distribution of GFP with that of other zonated proteins in the liver of *Cldn2-EGFP* mice harvested at different ages using double-immunofluorescence analysis. As previously shown (*Burke et al., 2018*; *Notenboom et al., 1997*), GS was expressed exclusively in the first 2–3 layers of hepatocytes surrounding the central veins at E18.5 (*Figure 1C*) and this restricted expression pattern remained unchanged at P2, P15 and P30 (*Figure 1C*). Similarly, Cyp2e1 expression was confined to a large area of pericentral/perivenous hepatocytes from E18.5 to adulthood (*Figure 1C*). Different to those proteins, claudin-2/GFP expression was not detected in hepatocytes at E18.5 (*Figure 1C*) whereas that protein was broadly expressed in hepatocytes at P2 (*Figure 1C* and *Figure 1—figure supplement 2B*). The wide distribution of claudin-2/GFP in hepatocytes was transient since this protein was largely confined to pericentral (GS$^+$) and perivenous (Cyp2e1$^+$) hepatocytes at both P15 and P30 (*Figure 1C,D*). On the other hand, we uncovered that E-cadherin was expressed in all hepatocytes at both E18.5 and P2 (*Figure 1C*; *Figure 1—figure supplement 2*) and was restricted to periportal hepatocytes at P15 and P30 (*Figure 1C*). Those findings concur with a published study (*Rocha et al., 2015*) showing that E-cadherin is not fully zonated in

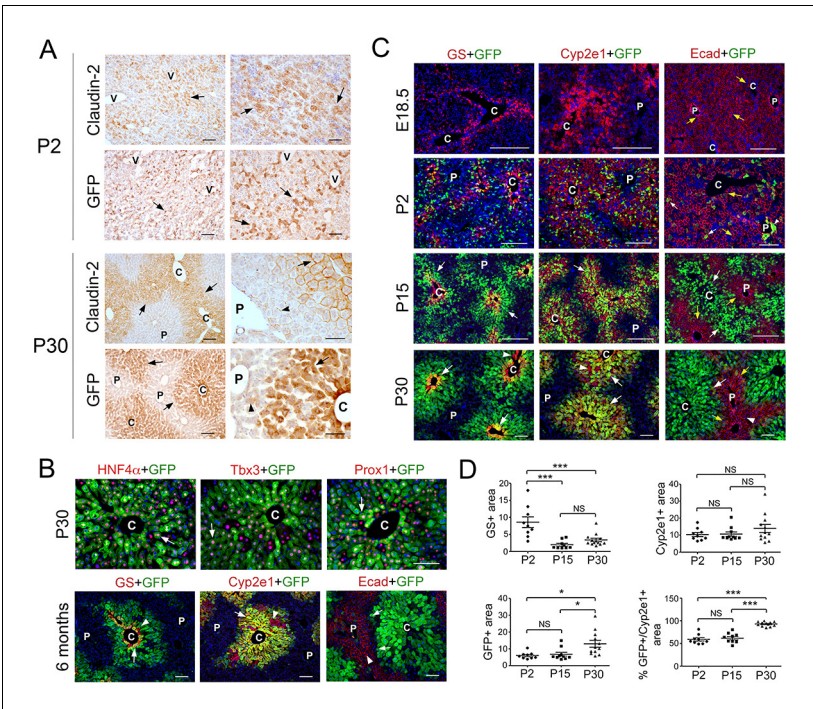

**Figure 1.** Claudin-2 expression becomes zonated during liver maturation. (**A**) Immunohistochemistry analysis of claudin-2 and claudin-2/GFP expression showing identical distribution of these proteins in hepatocytes (arrows) at P2 and P30. Few hepatocytes near the portal veins (arrowheads) also express low claudin-2 or claudin-2/GFP at P30. Wildtype specimens were used for claudin-2 analysis and *Cldn2-EGFP* specimens for claudin-2/GFP analysis. Scale bars: 50 μm (right); and 100 μm (left). (**B**) Double-immunofluorescence analysis showing co-expression of claudin-2/GFP (arrows) with the hepatocyte transcription factors HNF4α, Tbx3 and Prox1 around the central veins in P30 *Cldn2-EGFP* livers. Scale bars: 100 μm. (**C**) Double-immunofluorescence results show lack of claudin-2/GFP hepatic expression late in gestation (E18.5), numerous claudin-2/GFP$^+$ hepatocytes (arrows) distributed throughout the liver after birth (P2), and high claudin-2/GFP expression (arrows) restricted to Zone 3 (GS$^+$) and Zone 2/3 (Cyp2e1$^+$) hepatocytes in juvenile (P15) and adult (P30) livers (a few pericentral/perivenous hepatocytes [arrowheads] do not express GFP at P30). E-cadherin expression (yellow arrows) is detected in all hepatocytes at E18.5 and P2, and is restricted to periportal hepatocytes that lack high claudin-2/GFP expression at P15-P30. Scale bars: 200 μm (E18.5, P2, P15) and 100 μm (P30). (**D**) Quantification of GS$^+$, Cyp2e1$^+$ and claudin-2/GFP$^+$ areas and the relative abundance of claudin-2/GFP$^+$ hepatocytes in Zone 2 in postnatal (P2), juvenile (P15) and adult (P30) livers. 3–4 representative fields from three individual livers of each genotype were used for quantification. Statistical difference was determined by one-way ANOVA with Bonferroni's multiple comparisons test. *NS*, not significant (p>0.05), *p<0.05, ***p<0.001. (**A–C**) Each image represents 3–4 individual livers. (**C**), central vein. (P), portal vein. (V), vein. Related data can be found in *Figure 1—figure supplements 1–3*.

The online version of this article includes the following source data and figure supplement(s) for figure 1:

**Source data 1.** Quantification of GS+, Cyp2e1+ and claudin-2/GFP+ areas and the relative abundance of claudin-2/GFP+ hepatocytes in Zone two in P2, P15 and P30 livers.

**Figure supplement 1.** Claudin-2 and GFP proteins have identical expression in intrahepatic and extrahepatic biliary tissues of wildtype and *Cldn2-EGFP* mice.

**Figure supplement 2.** E-cadherin and claudin-2/GFP are expressed throughout the newborn liver.

**Figure supplement 3.** PEPCK expression is not restricted to periportal hepatocytes at birth.

---

the murine liver at P10. Similar to E-cadherin, the periportal enzyme phosphoenolpyruvate carboxy-kinase (PEPCK) was broadly expressed at birth (P0) and was restricted to Zone one by P11 (*Figure 1—figure supplement 3A*), and this transition coincided with induction of the Wnt/β-catenin inhibitor APC (*Benhamouche et al., 2006*) in periportal hepatocytes (*Figure 1—figure supplement 3C*). In conclusion, our study uncovered a dynamic, non-synchronous spatiotemporal segregation of metabolic and membrane-associated hepatocyte functions in the maturing liver.

# Endothelial-secreted Wnt ligands maintain claudin-2 zonated expression in the adult liver

Wnt/β-catenin signaling is a key regulator of metabolic zonation (*Benhamouche et al., 2006*; *Burke et al., 2009*; *Planas-Paz et al., 2016*; *Sekine et al., 2006*; *Yang et al., 2014*) and recent studies demonstrated that hepatic endothelial cells (HECs) produce Wnt ligands and agonists that preserve homeostatic metabolic zonation in the adult liver (*Leibing et al., 2018*; *Planas-Paz et al., 2016*; *Preziosi et al., 2018*; *Rocha et al., 2015*; *Wang et al., 2015*). We observed that most claudin-2/GFP[+] hepatocytes juxtaposed HECs in the liver of *Cldn2-EGFP* newborn and adult mice (*Figure 2A*) and this result suggested to us that claudin-2 zonated expression requires endothelial Wnt signaling. To investigate this possibility, we deleted the *Wls* gene (encoding a protein necessary for Wnt exocytosis) (*Carpenter et al., 2010*) in murine ECs using conditional-deletion approaches.

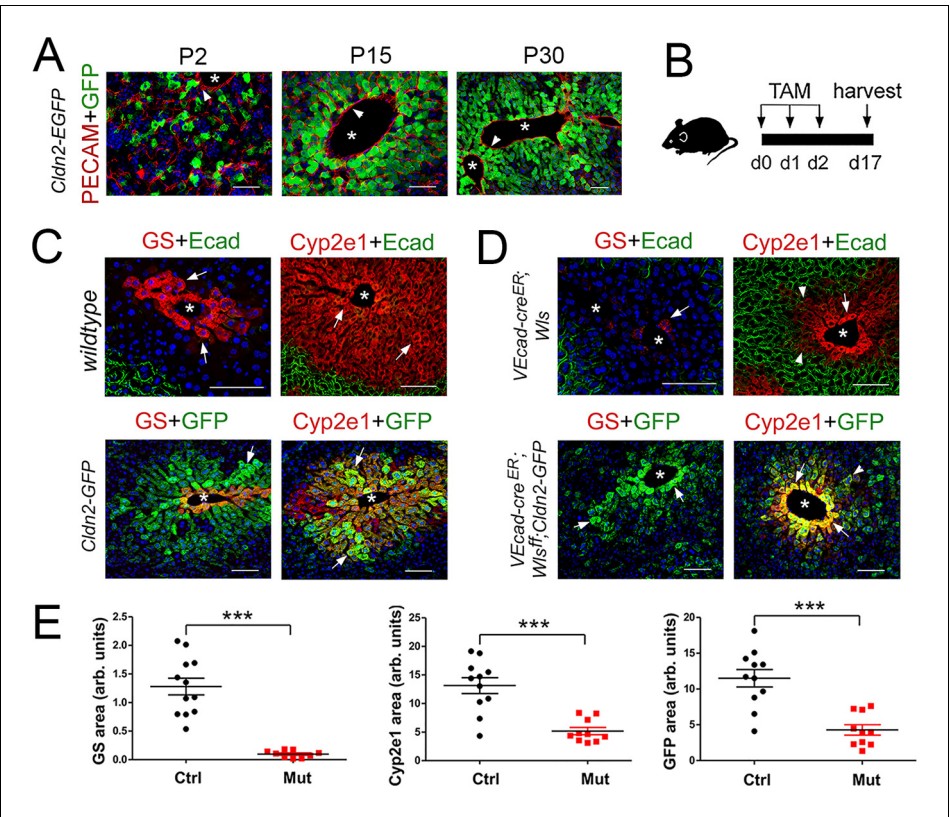

**Figure 2.** *Wls* deletion in HECs severely reduces the expression of claudin-2 and Cyp2e1 in perivenous hepatocytes in adult livers. (**A**) Immunofluorescence results showing that pericentral claudin-2/GFP[+] hepatocytes physically contact the central vein endothelium (PECAM[+], arrowheads) in newborn (P2) juvenile (P15) and adult (P30) *Cldn2-EGFP* livers. (**B**) Schematic of tamoxifen administration and tissue harvesting. (**C**) Tamoxifen injection does not affect the zonated distribution of GS, Cyp2e1 and E-cadherin in wildtype livers (*top*, arrows), or claudin-2/GFP, GS and Cyp2e1 (*bottom*, arrows) in *Cldn2-GFP* livers. (**D**) *Wls* deletion in endothelial cells causes near depletion of GS[+] hepatocytes (*top left,* arrows), decreases both Cyp2e1 (*right* panels, arrows) and claudin-2/GFP perivenous expression (*bottom* panels, arrows; arrowheads show distal perivenous hepatocytes expressing low GFP), and expands E-cadherin expression into the margins of Zone 2 (*top right*, arrowheads). (**E**) Quantification of Zone 3 (GS[+]) and Zone 2 (Cyp2e1[+]/GFP[+]) areas in adult livers with or without endothelial *Wls*. Statistical difference was determined by two-tailed unpaired Student's *t*-test (***p<0.001, 3–4 representative fields from three individual livers of each genotype were used for quantification). Each image represents three individual livers. Asterisks indicate central vein lumens. Scale bars: 50 μm (**A**), 100 μm (**C,D**). Related data can be found in *Figure 2—figure supplement 1*.

The online version of this article includes the following source data and figure supplement(s) for figure 2:

**Source data 1.** Quantification of Zone three and Zone two areas in adult livers with or without endothelial Wls.
**Figure supplement 1.** *Wls* deletion in adult HECs disrupts zonated protein expression in hepatocytes.

Adult wildtype (control) mice, *Cldn2-EGFP* (control) mice, *Cdh5-CreERT2;Wls^f/f* (*Wang et al., 2010*) (*Wls*-deleted) mice, and *Cdh5-CreERT2;Wls^f/f;Cldn2-EGFP* (*Wls*-deleted) mice were injected with tamoxifen and their liver was harvested and processed for immunostaining 2 weeks later (*Figure 2B*). Tamoxifen treatment did not affect GS, Cyp2e1 or E-cadherin expression in wildtype livers (*Figure 2C*), or claudin-2/GFP expression in *Cldn2-EGFP* livers (*Figure 2C* and data not shown). In contrast, *Wls* endothelial deletion eliminated GS pericentral expression almost entirely (*Figure 2D,E*; *Figure 2—figure supplement 1A,B*) and reduced Cyp2e1 perivenous expression significantly (*Figure 2D,E*; *Figure 2—figure supplement 1A,C*) in *Cdh5-CreERT2;Wls^f/f* and *Cdh5-CreERT2;Wls^f/f;Cldn2-EGFP* livers. Also, *Wls* endothelial deletion reduced claudin-2/GFP expression (*Figure 2D,E*) and claudin-2 expression (*Figure 2—figure supplement 1A,D*) significantly in perivenous areas of *Cdh5-CreERT2;Wls^f/f;Cldn2-EGFP* and *Cdh5-CreERT2;Wls^f/f* livers. Furthermore, *Wls* endothelial deletion caused ectopic expansion of E-cadherin expression in perivenous hepatocytes of *Cdh5-CreERT2;Wls^f/f* and *Cdh5-CreERT2;Wls^f/f;Cldn2-EGFP* livers (*Figure 2D* and data not shown). These results demonstrate that endothelial Wnt signaling preserves metabolic and non-metabolic zonation in the adult murine liver.

## LSEC Wnt ligand secretion is dispensable to initially establish hepatic zonation and is necessary to maintain adult hepatic zonation

Murine HECs are heterogeneous (*Halpern et al., 2018*) and include both central vein endothelial cells (CEVs) and liver sinusoidal endothelial cells (LSECs). CEVs and LSECs express CD31/PECAM-1 whereas LSECs uniquely express the lymphatic marker Lyve-1 (*Halpern et al., 2018*; *Mouta Carreira et al., 2001*). Also, recent studies showed that CEVs express Wnt2 and Wnt9b ligands and the Wnt agonist R-spondin 3 (Rspo3) and LSECs express Wnt2 ligands in the homeostatic adult mouse liver (*Ding et al., 2010*; *Halpern et al., 2018*; *Preziosi et al., 2018*; *Rocha et al., 2015*; *Wang et al., 2015*; *Zhao et al., 2019*). Similar to those findings, we detected *Wnt2, Wnt9b* and *Rspo3* transcripts in cells lining the central veins in adult (P60) and newborn (P2) mouse livers (*Figure 3A*). We also uncovered that LSECs adjacent or close to the central veins expressed *Wnt2, Wnt9b* and *Rspo3* in P2 livers (*Figure 3A*) whereas LSECs located more distal to the central veins expressed *Wnt2* in P60 livers (*Figure 3A* and *Figure 3—figure supplement 1*). Additional results of immunofluorescence analysis showed that most GS^+ hepatocytes in Zone 3 and most Cyp2e1^+ hepatocytes in Zone 2/3 adjoined LSECs in the liver of newborn (P2), juvenile (P15) and adult (P30) mice (*Figure 3B*). Similarly, most claudin-2/GFP^+ hepatocytes were contiguous to LSECs in *Cldn2-EGFP* livers at P2, P15 and P30 (*Figure 3B*). These findings posited that Zone 2/3 hepatocyte expression requires Wnt ligands secreted from LSECs. To test this notion, we deleted *Wls* in LSECs of *Cldn2-EGFP* mice using *Lyve1-Cre* mice (*Pham et al., 2010*).

First, we performed lineage tracing experiments in *Lyve1-Cre;ROSA-LacZ* mice and found that LSECs (Lyve1^+) expressed the β-galactosidase reporter protein and CEVs (PECAM^+/Lyve1^-) lacked that protein (*Figure 3—figure supplement 2*). These results demonstrated that Wnt ligand secretion should be preserved in CEVs and abolished in LSECs using *Lyve1-Cre*. Next, we produced *Lyve1-Cre;Cldn2-EGFP* mice (control) and *Lyve1-Cre;Wls^f/f;Cldn2-EGFP* mice (*Wls*-deficient) and analyzed their liver using immunofluorescence, morphometry and immunohistochemistry analyses. We uncovered slightly reduced abundance of GS^+ hepatocytes and normal abundance of Cyp2e1^+, claudin-2/GFP^+ and E-cadherin^+ hepatocytes (*Figure 3C*) in *Lyve1-Cre;Wls^f/f;Cldn2-EGFP* livers compared to control livers at P2. In contrast, GS expression was almost undetected and the areas of Cyp2e1 and claudin-2/GFP expression were significantly reduced and restricted to a few layers of pericentral hepatocytes in *Lyve1-Cre;Wls^f/f;Cldn2-EGFP* adult (P30) livers in comparison to control livers (*Figure 3C*). Identical alterations in GS, Cyp2e1 and claudin-2 hepatocyte expression were also noticed in the liver of *Lyve1-Cre;Wls^f/f* adult mice (*Figure 3—figure supplement 3*). Interestingly, *Wls* deletion in LSECs also affected E-cadherin expression since we observed ectopic expression of this protein in pericentral and perivenous hepatocytes in *Lyve1-Cre;Wls^f/f;Cldn2-EGFP* adult livers (*Figure 3C*). Results from qRT-PCR experiments confirmed significant decreases in pericentral/perivenous transcripts (i.e., *Glul, Cyp2e1, Axin2, Cldn2* and *Tbx3*), and significant upregulation of the periportal transcripts *Apc* and *Cdh1* (*Figure 3D*) in P30 *Lyve1-Cre;Wls^f/f* livers. In summary, we determined that Wnt ligand secretion from LSECs is dispensable or redundant for the initiation of hepatic zonation, and is necessary to maintain metabolic and non-metabolic zonation in the adult liver.

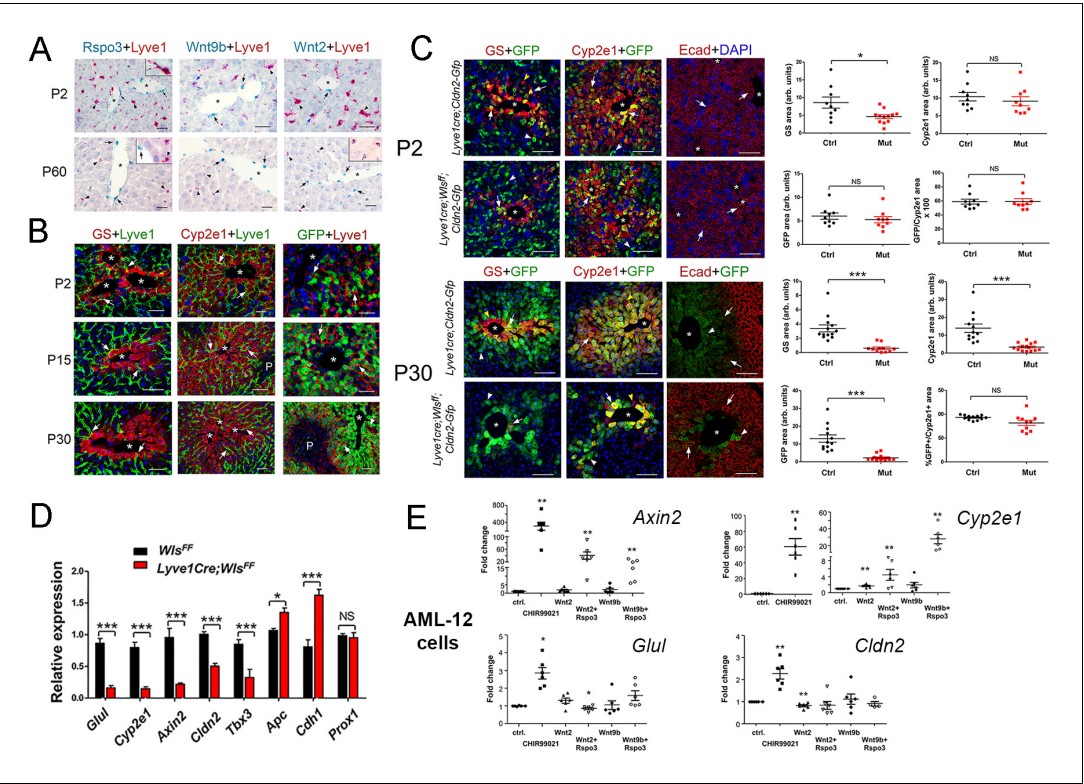

**Figure 3.** Lack of Wnt ligand secretion from LSECs impairs adult zonation maintenance. (A) Double in situ hybridization for *Rspo3* (green), *Wnt9b* (green) and *Wnt2* (green) showing a few LSCEs (Lyve1$^+$, red) expressing those transcripts (blue arrows and inset) in P2 livers. Only *Wnt2* transcripts are detected in some LSECs (inset) in P60 livers. Arrows indicate central vein endothelial cells and arrowheads indicate LSECs. Scale bars: 25 μm. Each image is representative of 3 individual mice (*n* = 3). (B) Double-immunofluorescence results show that hepatic Zones 3 (GS$^+$) and 2/3 (Cyp2e1$^+$) are densely irrigated by the hepatic sinusoids (Lyve1$^+$, arrows) in P2, P15 and P30 wildtype livers. The sinusoidal vasculature (arrows) is also in direct contact with claudin-2/GFP$^+$ hepatocytes in P2, P15 and P30 *Cldn2-EGFP* livers. Scale bars: 50 μm. Each image is representative of 2–4 individual mice (*n* = 2–4). (C) P2, Quantitative double immunofluorescence results show that Zone 2 (Cyp2e1$^+$, claudin-2/GFP$^+$) is relatively unchanged and Zone 3 (GS$^+$) is significantly reduced in *Lyve1-Cre;Wls$^{f/f}$;Cldn2-GFP* livers at P2. E-cadherin expression (arrows) is indistinguishable in P2 livers with or without endothelial *Wls*-deletion. P30, Similar quantitative results demonstrate that GS$^+$ hepatocytes are nearly absent, Zone 2 (Cyp2e1$^+$/GFP$^+$) is significantly reduced and restricted to pericentral areas, and E-cadherin expression (arrows, arrowheads are GFP$^+$ hepatocytes) is expanded towards the central veins in P30 *Lyve1-Cre;Wls$^{f/f}$;Cldn2-GFP* livers. 3–4 representative fields from three individual livers of each genotype were used for quantification. *p* values were determined by two-tailed unpaired Student's *t*-test, *NS*, not significant (p>0.05), *p<0.05, ***p<0.001. Arrows indicate GFP-double positive hepatocytes, white arrowheads are GFP$^+$ hepatocytes and yellow arrowheads are GFP$^-$ hepatocytes. Scale bars: 100 μm (D) Q-PCR results demonstrate reduced expression of Zone 2/3 transcripts, increased expression of Zone 1 transcripts, and normal levels of the hepatocyte transcript *Prox1*, in adult *Lyve1-Cre;Wls$^{f/f}$* livers (*n* = 3). *p* values were determined by two-way ANOVA, *NS*, not significant (p>0.05), *p<0.05, ***p<0.001. (E) Q-PCR results showing the effects of culturing AML-12 mouse hepatic cells with CHIR99021, Wnt2, Wntb9, or Wnt2/Wnt9b plus Rspo3 on *Axin2, Cyp2e1, Glul* and *Cldn2* expression. *p* values from two-tailed unpaired Student's t-test, *p<0.05, ***p<0.01; n = 6. (A–C) Asterisks indicate central vein lumens. Related data can be found in *Figure 3—figure supplements 1–3*.

The online version of this article includes the following source data and figure supplement(s) for figure 3:

**Source data 1.** Quantification of GS+, Cyp2e1+ and claudin-2/GFP+ areas and the relative abundance of claudin-2/GFP+ hepatocytes in P2 and P30 *Lyve1-Cre;Wls$^{f/f}$;Cldn2-GFP* livers, and Quantification of Wnt/β-catenin target genes expression of P30 *Lyve1-Cre;Wls$^{f/f}$;Cldn2-GFP* livers.

**Figure supplement 1.** The *Lyve-1* probe used for in situ hybridization in LSECs stains lymphatic endothelial cells.

**Figure supplement 2.** Lineage tracing results demonstrate selective β-gal expression in LSECs in *Lyve1-Cre; ROSA-LacZ* livers.

**Figure supplement 3.** *Wls* deletion in the adult endothelium disrupts liver zonation.

To further examine the effects of Wnt/β-catenin stimulation on claudin-2 hepatocyte expression, we cultured the murine hepatocyte-derived cell line AML12 (*Lehwald et al., 2011*) in the presence of CHIR99021 (a GSK-3 inhibitor that promotes nuclear β-catenin accumulation (*Gerbal-Chaloin et al., 2014*), Wnt2, Wnt9b, or Wnt2/Wnt9b and Rspo3. QRT-PCR results showed significantly increased expression of the Wnt/β−catenin target *Axin2* in AML12 cells incubated with CHIR99021, Wnt2+Rspo3 or Wnt9b+Rspo3 (*Figure 3E*). Similarly, CHIR99021, Wnt2 and Wnt9b increased *Cyp2e1* expression in those cells and the addition of Rspo3 further enhanced the effect of Wnt2/Wnt9b (*Figure 3E*). On the other hand, although CHIR99021 stimulated *Glul* and *Cldn2* expression by 2–3–fold (*Figure 3E*) the presence of Wnt2/Wnt9b alone or in combination with Rspo3 did not increase those transcripts (in fact, Wnt2 marginally reduced their levels; *Figure 3E*). The reason(s) behind the distinct responses of *Cyp2e1*, *Glul* and *Cldn2* to Wnt signaling are unclear but could involve variable thresholds for Wnt stimulation, synergy of Wnt/β-catenin signaling with other pathways, and/or differences in cellular composition between AML-12 cells and intact hepatocytes.

## Zones 3 and 2 are reestablished differently in the regenerating liver after acute injury

In rodents administered a CCl$_4$ bolus, the cytochrome P450 enzyme Cyp2e1 metabolizes this compound into highly reactive free radicals that cause cellular damage and necrosis of Zone 2/3 hepatocytes. This process is followed by a regenerative phase encompassing hepatocyte mass recovery, restoration of zonated patterns, and remodeling of the entire organ's architecture (*Tanaka and Miyajima, 2016*). In spite of the extensive literature on the effects of CCl$_4$, our knowledge about the reestablishment of hepatic zonation following acute or chronic exposure to CCl$_4$ is still limited and most publications in this topic did not examine parenchymal and non-parenchymal cell behaviors simultaneously (*Font-Burgada et al., 2015*; *Pu et al., 2016*; *Zhao et al., 2019*). Furthermore, we do not know if claudin-2 expression is affected following acute CCl$_4$ exposure or how the expression of this protein is restored in the injury model. To address those issues, we injected *Cldn2-EGFP* mice with CCl$_4$ and analyzed their liver 2–7 days post-injection (*Figure 4A*). Quantification of ALT/AST serum levels validated the occurrence of liver damage at day two and functional recovery of this organ at day 7 (*Figure 4B*) after a CCl$_4$ bolus.

To examine changes in zonation patterns downstream of CCl$_4$ acute exposure, we stained *Cldn2-EGFP* livers with anti-GS (Zone 3), anti-Cyp2e1 (Zone 2/3), anti-E-cadherin (Zone 1), anti-GFP (claudin-2$^+$ hepatocytes), and anti-Tbx3 (Zone 2/3) antibodies. In comparison to control livers, (*Figure 4C*) most hepatocytes in Zone 3 (GS$^+$) looked enucleated and necrotic and only 1–2 layers of normal-looking nucleated Cyp2e1$^+$/GFP$^+$ hepatocytes remained in Zone 2 (*Figure 4D*) 2 days post-CCl$_4$ injection. On the other hand, a few nucleated GS$^+$ hepatocytes reappeared at the margins of an enlarged Cyp2e1$^+$/GFP$^+$ area and their number increased 3–5 days post-CCl$_4$ administration (*Figure 4D*). Finally, the areas of GS, Cyp2e1 and claudin-2/GFP expression looked nearly restored 7 days after the initial CCl$_4$ bolus (*Figure 4D*). The analysis of temporal patterns of tissue damage and zonation recovery in the liver of wildtype mice injected with CCl$_4$ showed identical results (including claudin-2 expression; *Figure 4—figure supplement 1*).

Our analysis revealed that E-cadherin hepatic expression also changes upon CCl$_4$ exposure since Zone 2/3 GFP$^+$ hepatocytes did not express this protein in control livers (*Figure 4C*) and most GFP$^+$ hepatocytes expressed E-cadherin 2–5 days after CCl$_4$ injection (*Figure 4D*). Thus, in the CCl$_4$-acutely injured liver E-cadherin expression transiently expands into Zone two during the 'injury phase' and then retracts to its original periportal (Zone 1) domain during the 'repair phase'. Finally, since Tbx3 and claudin-2/GFP proteins colocalize extensively in the uninjured adult liver (*Figures 1C* and *4C*) we compared their expression in the *Cldn2-EGFP* injured liver and uncovered broad overlap of Tbx3 and GFP immunoreactivity throughout the course of recovery (i.e., 2–7 days post-CCl$_4$; *Figure 4C*). More interesting, we observed that GFP$^−$ periportal hepatocytes transiently expressed low-to-moderate levels of Tbx3 2–5 days post-CCl$_4$ injection (*Figure 4C*). This result argues that periportal hepatocytes also respond to local cues during the 'repair phase' that follows CCl$_4$-induced injury.

To further investigate how Zone 2/3 (Cyp2e1$^+$) hepatocytes recover in the CCl$_4$-injured liver, we injected wildtype mice with ethynyldeoxyuridine (EdU) 2 days after the initial CCl$_4$ bolus and sacrificed the animals 2 hr or 5 days (i.e., 120 hr) later. Results of quantitative immunofluorescence

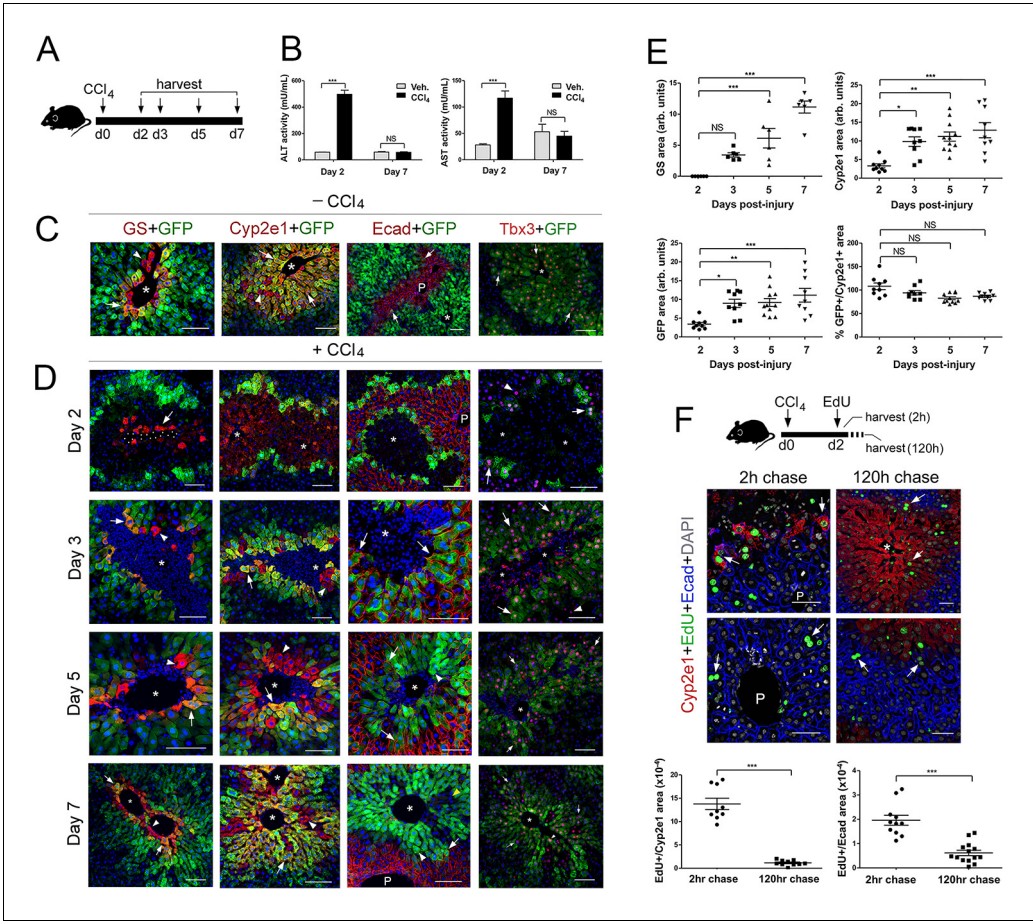

**Figure 4.** Proliferation and not de novo expression restore Claudin-2[+] and Cyp2e1[+] hepatocytes in the $CCl_4$-injured liver. (**A**) Experimental strategy for *Cldn2-EGFP* mice injected with $CCl_4$. (**B**) ALT/AST serum levels demonstrate liver damage 2 days post-$CCl_4$ (*n* = 3). *p* values were determined by two-way ANOVA, *NS*, not significant (p>0.05), ***p<0.001. (**C**) Expression of Zone 3 (GS[+]), Zone 2/3 (Cyp2e1[+], claudin-2/GFP[+]) and Zone 1 (Ecad[+]) markers and Tbx3 in the corn oil-injected *Cldn2-EGFP* adult liver. (**D**) Spatial distribution of the markers indicated above during the recovery phase that follows a $CCl_4$ bolus (see text for details). (*Day 2*, arrow, enucleated GS[+] cell. *Days 3–7*, arrows, GFP double-positive hepatocytes; arrowheads, GFP[−] hepatocytes. Asterisks indicate central vein lumens.) (**E**) Quantification of GS[+], Cyp2e1[+] and claudin-2/GFP[+] areas, and the GFP[+]/Cyp2e1[+] ratio, 2–7 days post-$CCl_4$. 3–4 representative fields from three individual livers dissected at the indicated time points were used for quantification. *p* values were determined by one-way ANOVA with Bonferroni's multiple comparisons test, *NS*, not significant (p>0.05), *p<0.05, **p<0.01, ***p<0.001. (**F**) (*Top*) Schematic of EdU administration and tissue harvesting post-$CCl_4$. (*Middle*) Immunofluorescence results show EdU incorporation (arrows) in Cyp2e1[+] hepatocytes located in the Zone 2 remnant and the undamaged Zone 1 (Ecad[+]) 2 days post-$CCl_4$. Both Cyp2e1[+] hepatocytes (arrows, *top*) and Ecad[+] hepatocytes located at the margins of restored Zone 2 (arrows, *bottom*) retain the EdU label after a 5 day chase. (*Bottom*) Quantification of EdU[+] hepatocytes in Zone 2/3 and Zone 1. 3–4 representative fields from three individual livers dissected at the indicated time points were used for quantification. *p* values were determined by two-tailed unpaired Student's *t*-test, ***p<0.001. (**C,D,F**) asterisks and dots indicate central vein lumens and images are from 3 to 4 individual livers. (**B, F**) *n* = 3. *NS*, not significant (p>0.05), *p<0.05, **p<0.01, ***p<0.001. P, portal veins. Scale bars: 100 μm (**C, D**) and 50 μm (**F**). Related data can be found in *Figure 4—figure supplements 1* and *2*.

The online version of this article includes the following source data and figure supplement(s) for figure 4:

**Source data 1.** Quantitation of ALT, AST, zonal markers and EdU[+] hepatocytes in $CCl_4$-injured liver.
**Figure supplement 1.** Changes in liver histology and Zone 2/3 protein expression after a $CCl_4$ bolus.
**Figure supplement 2.** Sox9[+] hepatocytes do not contribute to Zone 2/3 restoration after $CCl_4$ induced acute liver injury.

analysis showed incorporation of EdU in the surviving Cyp2e1$^+$ hepatocytes after a 2 hr chase (*Figure 4F*) and persistent EdU labeling in the expanded Zone 2 (Cyp2e1$^+$) after a 5 day chase (*Figure 4F*). In addition, we detected EdU incorporation in Ecad$^+$/Cyp2e1$^-$ periportal hepatocytes after a 2 hr or 5 day chase (*Figure 4F*). This result was intriguing because a recent study (*Font-Burgada et al., 2015*) showed restoration of hepatocyte mass from Sox9$^+$ periportal hepatocytes in a mouse model of chronic liver damage. Therefore, we investigated if Sox9$^+$-hepatocytes contribute to restore metabolic zonation in the CCl$_4$-injured liver by injecting *Sox9$^{CreER}$;ROSA-EGFP* mice with tamoxifen and then CCl$_4$ and harvesting the liver 7 days later (*Figure 4—figure supplement 2A*). Results of immunostaining showed expression of the GFP lineage-tracer in a few periportal hepatocytes and no GFP expression in Cyp2e1$^+$ hepatocytes (*Figure 4—figure supplement 2B*) in the CCl$_4$-injured *Sox9$^{CreER}$;ROSA-EGFP* liver. This finding ruled out the contribution of Sox9$^+$ periportal hepatocytes to metabolic Zone 2/3 restoration after CCl$_4$-acute damage. Instead, our results indicate that in this injury model metabolic Zone 2 is recovered via proliferation of undamaged hepatocytes and metabolic Zone 3 is induced de novo.

To complement our analysis of *Cldn2-GFP* livers exposed to CCl$_4$, we investigated the composition of the immune infiltrates in tissues harvested 2–7 days after CCl$_4$ injection (*Figure 5A*) using double-immunofluorescence and antibodies recognizing the pan-macrophage marker F4/80, the M1 macrophage marker CD86, and the M2 macrophage marker CD206 (*Shapouri-Moghaddam et al., 2018*). This analysis showed that most macrophages in *Cldn2-GFP* control livers were parenchymal Kupffer cells since they expressed F4/80 (*Figure 5B*) but not CD86 (*Figure 5C*) or CD206 (data not shown). On the other hand, in CCl$_4$-injected *Cldn2-GFP* livers most macrophages infiltrating the central veins represented M1 macrophages that expressed F4/80 and CD86 (*Figure 5B,C*) but not CD206 (*Figure 5D* and data not shown). Quantitative results demonstrated that the average abundance of perivenous macrophages increased at around day three post-CCl$_4$ and began to decline afterwards (*Figure 5E*). Also, most CD86$^+$ macrophages concentrated around the central veins in CCl$_4$-injected livers (*Figure 5C*) although in periportal areas of this organ a few isolated CD86$^+$ cells were also found within the hepatic sinusoids (*Figure 5F*). This finding argues that perhaps some periportal CD86$^+$ macrophages travel through the sinusoids towards pericentral areas in the CCl$_4$-damaged liver.

## Endothelial Wnts reestablish claudin-2 zonated expression in hepatocytes following CCl$_4$-induced acute injury

HECs produce Wnts and other factors that stimulate liver growth (*Ding et al., 2010*; *Hu et al., 2014*; *Leibing et al., 2018*; *Preziosi et al., 2018*; *Zhao et al., 2019*) and Wnt ligands that help to repair the architecture of the CCl$_4$-injured liver (*Zhao et al., 2019*). Our finding that most claudin2-GFP$^+$ hepatocytes were contiguous to HECs (PECAM-1$^+$) during the recovery period that follows a CCl$_4$ bolus (*Figure 6A*) posited that Wnt secretion from HECs help reestablishing metabolic and non-metabolic zonation in this injury model. To test this notion, we administered CCl$_4$ to *Cdh5-CreERT2* (control) and *Cdh5-CreERT2;Wls$^{f/f}$* mice at day 0, injected three consecutive daily doses of tamoxifen, and harvested the livers 2–15 days post CCl$_4$ injection. We uncovered similar increases in circulating ALT/AST levels at day 2 and numerous macrophage infiltrates in hepatic perivenous areas at day 5 in *Cdh5-CreERT2* and *Cdh5-CreERT2;Wls$^{f/f}$* mice (*Figure 6B*). These results validated the occurrence of hepatic damage in both mouse strains. Similar to our previous findings in *Cldn2-EGFP* mice (*Figure 2*), Zones 3 (GS$^+$) and 2 (Cyp2e1$^+$) looked almost recovered at day 7 and these areas were completely restored at day 15 in the *Cdh5-CreERT2* liver. In contrast, Zone 3 was nearly absent and Zone 2 was significantly reduced in *Cdh5-CreERT2;Wls$^{f/f}$* livers at days 5, 7 and 15 post-CCl$_4$ administration (*Figure 6B*). Also, E-cadherin expression retracted to its normal location in periportal Zone 1 in *Cdh5-CreERT2* livers 7–15 days post-CCl$_4$ injection whereas E-cadherin expression was ectopically expanded into perivenous areas in CCl$_4$-injured *Cdh5-CreERT2;Wls$^{f/f}$* livers dissected at similar time points (*Figure 6B*). These results demonstrate that endothelial Wnt secretion is necessary to reestablish Zone 3, to recover the normal size of Zone 2, and to restore E-cadherin expression to Zone 1 hepatocytes, in CCl$_4$-acutely injured livers.

HECs are heterogeneous and similar to hepatocytes these cells display zonated distribution of various proteins. For instance, recent studies in adult mouse livers showed that central vein endothelial cells express *Wnt2*, *Wnt9b* and *Rspo3* transcripts whereas perivenous sinusoidal cells express *Wnt2* transcripts (*Halpern et al., 2018*; *Zhao et al., 2019*). Also, *Kit* transcripts (encoding the surface

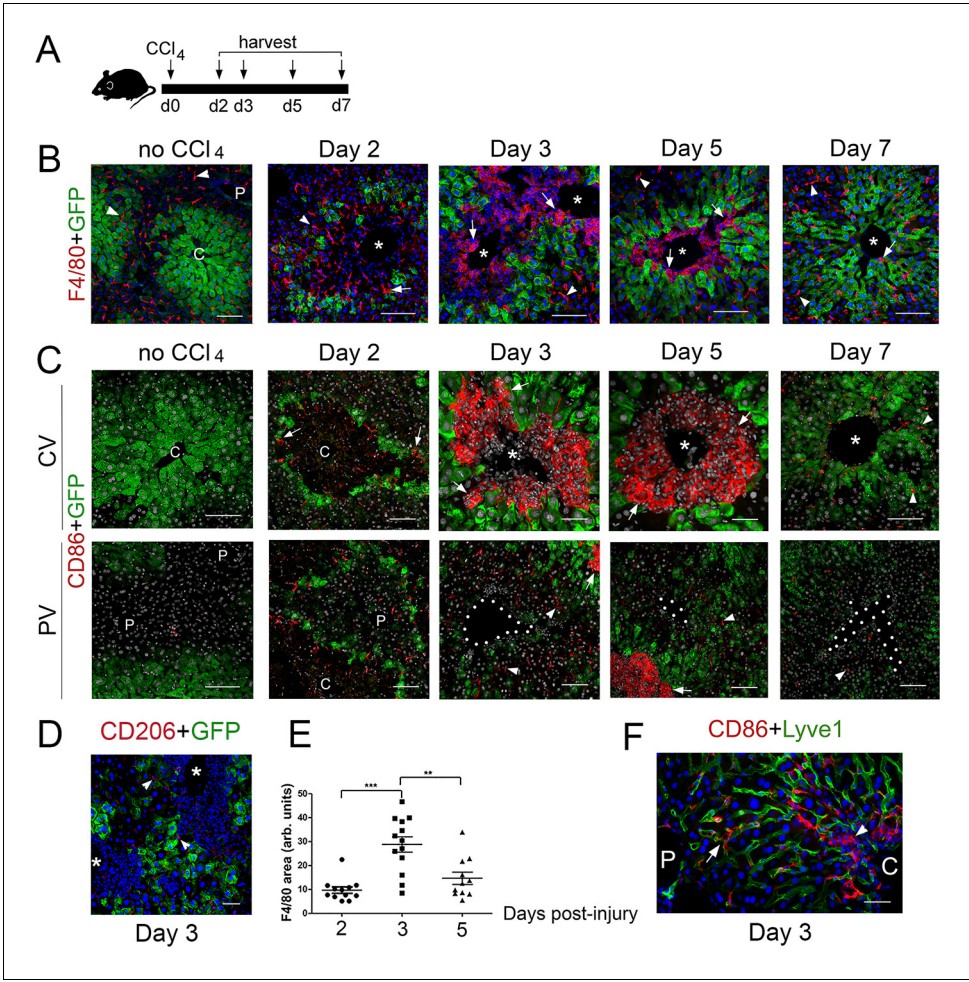

**Figure 5.** M1 macrophages infiltrate the perivenous space in CCl₄-acutely injured livers. (**A**) Experimental strategy for *Cldn2-EGFP* mice injected with CCl₄. (**B**) (*No CCl₄*) Results of double-immunofluorescence analysis show expression of the pan-macrophage marker F4/80⁺ in Kupffer cells (arrowheads) in a control adult liver (no CCl₄ injection). (*Day 2–5*) In the CCl₄-injected *Cldn2-EGFP* liver, macrophages (arrows) start to infiltrate the necrotic perivenous area at day two and form a physical barrier between the central vein endothelium and the expanding GFP⁺ Zone 2 at days 3 and 5. Resolution of macrophage perivenous infiltrates and restoration of claudin-2/GFP expression occur 7 days post-CCl₄ injection (arrows are Kupffer cells). (**C**) (*No CCl₄*) M1 macrophages (CD86⁺) are not detected in perivenous or periportal regions in *Cldn2-EGFP* control mice (*C* is central vein, *P* is portal vein). (*Day 2–5, top*) Immunofluorescence images of CCl₄-injected *Cldn2-EGFP* livers showing increasingly abundant infiltrates of CD86+ macrophages (arrows) 2 days post-CCl₄ and persistence of these cells around the central veins at days 3 and 5. CD86⁺ macrophages are no longer present in perivenous areas at day 7 (*CV*, central vein region). (*Day 2–5, bottom*) CD86⁺ macrophages are very scarce in periportal areas in CCl₄-injected livers (*PV*, portal vein region). (The exposure in the green channel was decreased for better visualization of the red [CD86] signal. (**D**) Macrophages infiltrating the perivenous region do not express the M2 macrophage marker CD206 3 days post-CCl4 injection (arrowheads indicate low expression of this marker in LSECs). (**E**) Quantification of F4/80⁺ immunofluorescence distribution 2-, 3- and 5 days post-CCl₄. *p* values were determined by one-way ANOVA with Bonferroni's multiple comparisons test. **$p < 0.01$, ***$p < 0.001$. 3–4 representative fields from three individual livers dissected at the indicated time points were used for quantification. Asterisks and *C*, central veins. *P*, portal veins. *CV*, central vein area. *PV*, portal vein area. (**F**) A few CD86⁺ macrophages are seen within the Lyve1⁺ hepatic sinusoids in periportal areas of the CCl₄ injected liver. CD86⁺ macrophages are overabundant in perivenous areas (right, 'C', arrowhead) in comparison to periportal areas ('P'). Each image represents 3–4 individual livers. Scale bars: 100 µm (**B**), (**C**) [no CCl₄, Days 2,7]) Scale bars: 50 µm (**C**) [Days 3,5], (**D,F**).

The online version of this article includes the following source data for figure 5:

**Source data 1.** Quantification of F4/80⁺ immunofluorescence distribution post-CCl₄ injection.

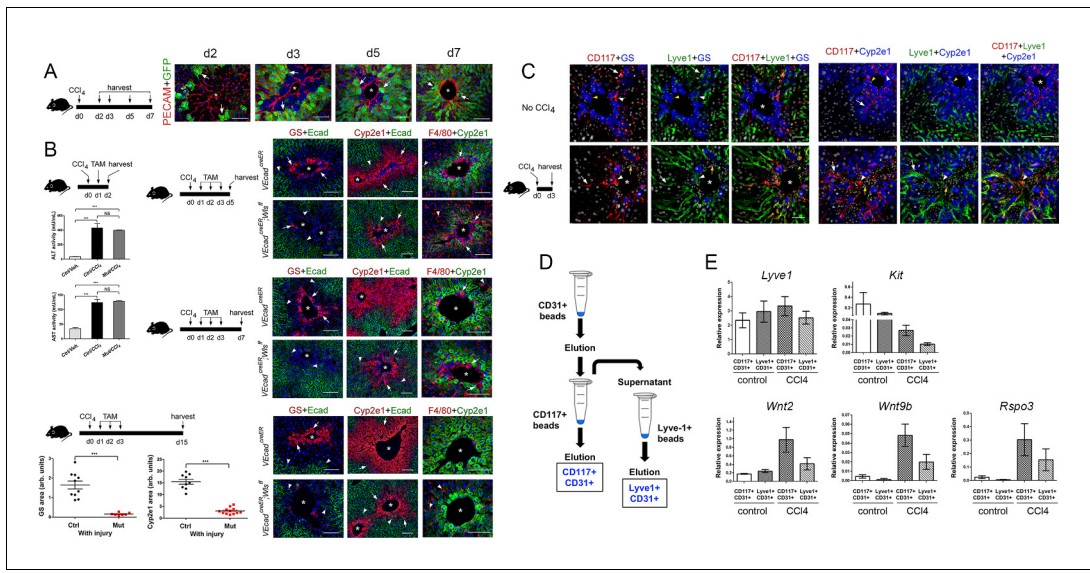

**Figure 6.** Endothelial Wnt ligand secretion reestablishes metabolic zonation in the $CCl_4$-acutely injured liver. (**A**) (*Left*) Schematic of $CCl_4$ administration and tissue harvesting using *Cldn2-EGFP* mice. (*Right*) Double-immunofluorescence results show physical association of claudin-2/GFP$^+$ hepatocytes (arrows) with hepatic endothelial cells (PECAM$^+$) throughout the recovery period that follows $CCl_4$-acute injury. (**B**) (*Top, left*) Schematic of the experimental strategy and tissue harvesting. ALT/AST serum levels demonstrate that $CCl_4$ promotes liver damage in *Cdh5-CreERT2* and *Cdh5-CreERT2;Wls$^{f/f}$* mice injected with tamoxifen. (*n* = 3). *p* values were determined by one-way ANOVA with Bonferroni's multiple comparisons test, *NS*, not significant (p>0.05), ***p<0.001. (*Right and bottom*) Diagrams indicate the experimental strategy and tissue harvesting. Double-immunofluorescence and quantitative results show progressive expansion and full restoration of Zone 3 (GS$^+$, arrows) and Zone 2 (Cyp2e1$^+$, arrows) and transient macrophage infiltrates (F4/80$^+$, arrows; arrowheads are Kupffer cells) in the liver of *Cdh5-CreERT2* mice after acute $CCl_4$ administration. In contrast, Zone 3 is nearly absent, Zone 2 is significantly smaller, and Zone 1 is expanded, in *Cdh5-CreERT2;Wls$^{f/f}$* livers 5-, 7- and 15 days post-$CCl_4$. Perivenous macrophage infiltrates are observed in both *Cdh5-CreERT2* and *Cdh5-CreERT2;Wls$^{f/f}$* livers 5–7 days post-$CCl_4$ but not 15 days after $CCl_4$ administration. *p* values were determined by two-tailed unpaired Student's *t*-test, ***p<0.001, (*n* = 3). (**C**) (*Left*) Schematic of $CCl_4$ administration and tissue harvesting. (*Right*) Triple-immunofluorescence results show that CD117 proteins are restricted to the Lyve-1$^+$/Lyve-1$^{LOW}$ sinusoidal endothelium traversing Zones 2/3 in the normal and $CCl_4$-injected (day 3) adult liver. (**D**) Schematic of CD117$^+$ and Lyve1$^+$ HECs isolation. Nonparenchymal liver cells (NPCs) were isolated using a two-step collagenase perfusion method and incubated with CD31-coated Dynabeads. The eluted CD31$^+$ (HEC) fraction was incubated with CD117-coated Dynabeads to isolate CD117$^+$CD31$^+$ ('pericentral/perivenous') HECs. The unbound fraction from this step was incubated with Lyve-1-coated Dynabeads to separate Lyve$^+$CD31$^+$ hepatic sinusoidal cells from other HECs. (**E**) QRT-PCR results show comparable *Lyve1* transcript expression in CD117$^+$/CD31$^+$ and Lyve1$^+$/CD31$^+$ isolates from saline-injected ('control') or $CCl_4$-injected (day 3) livers, lower *Kit* expression in LSECs from injured livers compared to control livers, and higher *Wnt2*, *Wnt9b* and *Rspo3* expression in LSECs from injured livers compared to control livers. (Three individual livers per condition were used to isolate LSECs.) (**A–C**) Each image represents 2–4 individual livers. Asterisks: central veins. (**B**) *NS*, not significant (p>0.05), ***p<0.001, (*n* = 3). Scale bars: 50 µm (**C**), 100 µm (**A, B**).

The online version of this article includes the following source data for figure 6:

**Source data 1.** Quantification of ALT/AST serum levels, GS, Cyp2e1 immunofluorescence in *Cdh5-CreERT2;Wlsf/f* livers post-$CCl_4$ injection.

marker CD117) are abundant in HECs located close to the CV and low or absent in HECs located in periportal areas of the adult liver (*Halpern et al., 2018*). Using immunostaining analysis, we uncovered that the hepatic sinusoids traversing Zones 2 and 3 express CD117 but not those located in periportal liver areas of the adult mouse liver (*Figure 6C*). Also, in those tissues the CD117$^+$ sinusoidal cells located in pericentral areas expressed very low levels of Lyve-1 (*Figure 6C*). This expression pattern was almost identical in the $CCl_4$-injected liver (at day 3) although in the injured organ the LSECs within Zone 2 (Cyp2e1$^+$) showed more extensive colocalization of CD117 and Lyve-1 compared to the uninjured liver (*Figure 6C*). Since CD117 is expressed more abundantly in percentral/

perivenous LSECs than in periportal LSECs, we attempted to isolate these 2 cell populations using CD117-coated magnetic beads and then investigate their Wnt/Rspo expression profile under normal and acute injury conditions. For the 'injury' experiment, LSECs were isolated from mouse livers 3 days after a $CCl_4$ bolus because at this time point we observed de novo induction of GS expression, expansion of Zone 2 (Cyp2e1$^+$/claudin-2$^+$) and perivenous and periportal hepatocyte proliferation (*Figure 4*). Also, for these experiments we included an initial purification step with anti-CD31 antibodies to remove potential contamination of CD117$^+$ immune cells in the injured liver (*Figure 6D*). The results of qRT-PCR analysis showed relatively similar *Lyve1* expression between CD117$^+$/CD31$^+$ cells ('pericentral/perivenous LSECs') and Lyve-1$^+$/CD31$^+$ cells ('periportal LSECs'; *Figure 6E*). Also, this analysis showed relatively higher *Kit* expression in CD117$^+$/CD31$^+$ cells compared to Lyve-1$^+$/CD31$^+$ cells in both the homeostatic and injured liver. On the other hand, *Kit* expression was lower in LSECs from $CCl_4$-injured livers compared to LSECs from control livers (*Figure 6E*) and this result likely indicates that many perivenous LSECs were lost due to extensive tissue damage. Interestingly, the LSECs isolated from $CCl_4$-injured livers expressed higher levels of *Wnt2*, *Wnt9b* and *Rspo3* compared to LSECs from control livers, and this difference was more prominent in the CD117$^+$/CD31$^+$ LSEC fraction (*Figure 6E*). This result was surprising since *Rspo3* and *Wnt9b* transcripts are mainly expressed in pericentral LSECs in the uninjured liver (*Figure 3A* and *Zhao et al., 2019*). Therefore, we postulate that Wnt ligands and Wnt agonists are broadly upregulated in LSECs upon acute liver injury.

## *Lyve1-cre;Wls^{f/f}* mice are refractory to $CCl_4$-induced liver damage

As we found that the expanding claudin2-GFP$^+$ area continuously juxtaposed LSECs (Lyve-1$^+$) in *Cldn2-GFP* livers exposed to $CCl_4$ (*Figure 7A*) and pericentral/perivenous LSECs express Wnt ligands in the $CCl_4$-acutely injured liver (*Figure 6E*), we investigated whether *Wls* deletion from LSECs affects zonation restoration in the $CCl_4$-injury model. We used *Lyve1-Cre;Wls^{f/f}* mice for these experiments because we predicted that the remaining Cyp2e1$^+$ hepatocytes (*Figure 7B*) should be susceptible to $CCl_4$-induced toxicity. $CCl_4$ administration to *Lyve1-Cre* (control) mice led to extensive Zone 2 damage, loss of Zone 3 and elevated ALT/AST serum levels 2 days post injection (*Figure 7C*), extensive perivenous macrophage infiltration at day 4 (*Figure 7D*), and almost complete restoration of tissue architecture and metabolic zonation at day 7 (*Figure 7E*). Surprisingly, $CCl_4$ administration to *Lyve1-Cre;Wls^{f/f}* mice did not produce any obvious changes in tissue architecture or zonated marker distribution in comparison to *Lyve1-Cre;Wls^{f/f}* mice injected with saline (*Figure 7B–E*; *Figure 7—figure supplement 1*). Furthermore, *Lyve1-Cre;Wls^{f/f}* mice injected with $CCl_4$ had normal ALT/AST serum levels at day 2 (*Figure 7C*) and did not show perivenous macrophage infiltrates at day 4 (*Figure 7D*). We also injected *Cdh5-CreERT2;Wls^{f/f}* mice first with tamoxifen (to delete *Wls*) and 2 weeks later with $CCl_4$ (to induce Zone 2/3 hepatotoxicity) and then harvested their liver 3 days later. Similar to *Lyve1-Cre;Wls^{f/f}* mice, in *Cdh5-CreERT2;Wls^{f/f}* livers administered a $CCl_4$ bolus we did detect macrophage infiltrates (*Figure 7—figure supplement 2B*) or observed features indicative of liver damage (*Figure 7—figure supplement 2A*). Thus, we conclude that the remaining Cyp2e1$^+$ hepatocytes in *Lyve1-Cre;Wls^{f/f}* livers are protected or refractory to $CCl_4$-induced toxicity.

## Discussion

The concept of Liver Zonation was proposed almost 40 years ago to explain the spatial separation of metabolic functions in the adult liver (*Jungermann and Katz, 1989*). Subsequent breakthroughs were the identification of Wnt/β-catenin signaling as a key regulator of metabolic hepatic zonation (*Benhamouche et al., 2006*; *Burke et al., 2009*; *Burke et al., 2018*; *Planas-Paz et al., 2016*; *Sekine et al., 2006*; *Yang et al., 2014*) and the discovery that HECs are a major source of Wnt ligands in the adult liver (*Leibing et al., 2018*; *Planas-Paz et al., 2016*; *Preziosi et al., 2018*; *Rocha et al., 2015*; *Wang et al., 2015*). Our study extends those findings by showing that similar regulatory mechanisms control metabolic and non-metabolic zonation in the liver. Specifically, we demonstrate that: 1) endothelial Wnt signaling restricts the expression of a tight junction protein (claudin-2) and the expression of a cell adhesion protein (E-cadherin) to complementary regions in the adult liver; 2) the restoration of metabolic and non-metabolic zonation after $CCl_4$-acute injury requires Wnt ligands produced by endothelial cells; and 3) the secretion of Wnt ligands from hepatic

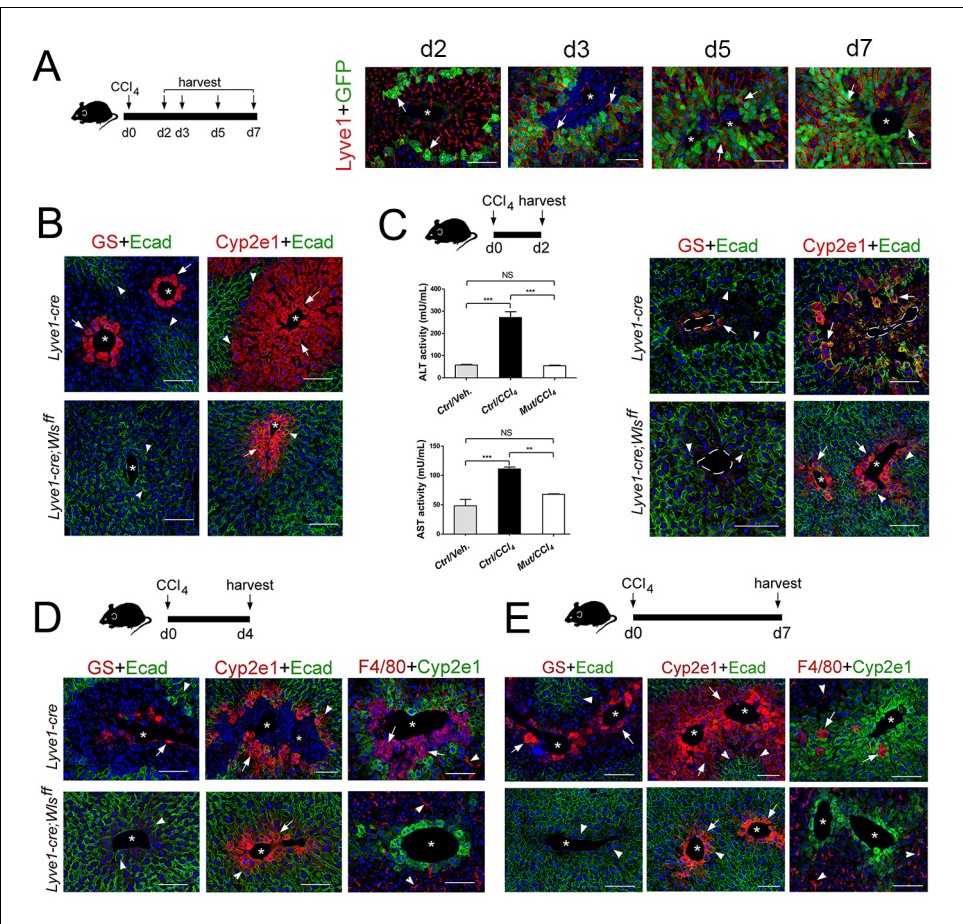

**Figure 7.** *Lyve1-Cre;Wls^f/f* mice are refractory to CCl₄-induced hepatotoxicity. (**A**) (*Left*) Schematic of CCl₄ administration and tissue harvesting using *Cldn2-EGFP* mice. (*Right*) Double-immunofluorescence results show that claudin-2/GFP⁺ hepatocytes (arrows) are in close proximity to the sinusoidal endothelium (Lyve1⁺, arrows) throughout the recovery period that follows CCl₄ acute injury. (**B**) GS (Zone 3), Cyp2e1 (Zone 2/3) and E-cadherin (Zone 1) expression in control and *Lyve1-Cre;Wls^f/f* livers without CCl₄ treatment. (**C–E**) *Top*: Schematics of tissue harvesting post-CCl₄ administration. (**C**) (*Left*) ALT/AST serum levels indicate liver damage in *Lyve1-Cre* (control) mice and no liver damage in *Lyve1-Cre;Wls^f/f* mice, 2 days post-CCl₄. *p* values were determined by one-way ANOVA with Bonferroni's multiple comparisons test. *NS*, not significant (p>0.05), **p<0.01, ***p<0.001 (*n* = 3). (**C–E**) (*top panels*): Immunostaining results show that in *Lyve1-Cre* livers injected with CCl₄, Zone 3 is nearly undetected (GS, arrows) and Zone 2 (Cyp2e1, arrows) is severely destroyed at day 2 (**C**); a few enucleated cells express GS (arrow) around the central vein, Zone 1 ('E-cad', arrowheads) is expanded, and Zone 2 ('Cyp2e1', arrow) is separated from the central vein by macrophage infiltrates ('F4/80', arrows; arrowheads are Kupffer cells) at day 4; and Zones 1–3 ('E-cad'/'Cyp2e1'/'GS', arrows) look nearly restored and macrophage infiltrates are scarce (arrows) at day 7. (**C–E**) (*bottom panels*): GS⁺ cells and macrophage infiltrates are undetected and both, Cyp2e1 and E-cadherin expression are unchanged, in *Lyve1-Cre;Wls^f/f* livers 2–7 days post-CCl₄. Each image represents 3–4 individual livers. Asterisks: central veins. Scale bars: 100 μm. Related data can be found in *Figure 7—figure supplements 1* and *2*.

The online version of this article includes the following source data and figure supplement(s) for figure 7:

**Source data 1.** Quantification of ALT/AST serum levels in in *Lyve1-Cre;Wls^f/f* mice post-CCl₄ injection.
**Figure supplement 1.** A CCl₄-bolus does not induce hepatotoxicity in *Lyve1-Cre;Wls^f/f* mice.
**Figure supplement 2.** *Wls* ablation using *VE-cadherin^creER* disrupts zonation and prevents CCl₄-hepatotoxicity.

---

sinusoidal cells is dispensable or redundant for the establishment of metabolic and non-metabolic zonation in the newborn liver, and is necessary to preserve zonation patterns in the adult liver.

Our study also introduces a novel transgenic mouse model (*Cldn2-EGFP*) suitable to investigate changes in claudin-2 expression during development, regeneration and disease. Using those mice,

we discovered a unique distribution of claudin-2 proteins in hepatocytes relative to other membrane proteins and enzymes both at birth and in adulthood. Claudin-2 function is necessary for proper bile flow and bile composition (*Matsumoto et al., 2014*; *Yeh et al., 2010*) and the spatiotemporal changes that we uncovered probably reflect how the maturing liver adapts to new metabolic demands and stressors. We also obtained evidence that the mechanism(s) that initiate claudin-2 expression in immature hepatocytes and the mechanism(s) that preserve claudin-2 expression in mature hepatocytes are different. Specifically, we found that Wnt ligand secretion from the hepatic sinusoidal endothelium is necessary to maintain claudin-2 expression in adult hepatocytes and dispensable for the onset of claudin-2 expression in newborn hepatocytes. Our discovery that Wnt/β-catenin signaling regulates claudin-2 expression in mature hepatocytes concurs with a published study showing that β-catenin deletion in adult hepatocytes causes significant downregulation of claudin-2 proteins (*Yeh et al., 2010*). Investigating how claudin-2 expression is regulated in hepatic cells has clinical relevance because deficiency of this protein causes intrahepatic cholestasis and increases gall stone formation in mice (*Matsumoto et al., 2014*; *Yeh et al., 2010*). To further investigate the effects of Wnt signaling on claudin-2 expression, we performed experiments in the hepatocyte-derived cell line AML-12. Surprisingly, the addition of Wnt2 or Wnt9b alone or in combination with Rspo3 had no demonstrable effect on *Cldn2* expression whereas those same factors stimulated the expression of *Axin2* (a known Wnt/β-catenin target gene) in AML-12 cells. Possible explanations for those unanticipated results include differences in the cellular context of immortalized AML-12 cells compared to intact hepatocytes, or the requirement of distinct Wnt thresholds and/or specific Wnt ligand combinations to stimulate *Axin2* vs. *Cldn2* hepatocyte expression.

Another interesting finding in our study is that preserving metabolic and non-metabolic zonation in the adult liver requires Wnt ligand secretion from LSECs. Using the same *Lyve1-Cre* mouse strain that we used here, other researchers produced *Lyve1-Cre;Wls$^{f/f}$* mice and similar to us they described that GS and Cyp2e1 proteins were absent or barely detected in the mutant adult liver (*Preziosi et al., 2018*). On the other hand, that study did not examine alterations in other zonated proteins in the mutant adult liver or the phenotype of *Lyve1-Cre;Wls$^{f/f}$* newborn livers. Furthermore, those authors concluded that *Wls* was deleted in both CEVs and LSECs in *Lyve1-Cre;Wls$^{f/f}$* livers. In contrast, our results of lineage tracing experiments invalidated that cre recombinase was ever expressed in CEVs in our *Lyve1-Cre;Wls$^{f/f}$* mice. Whether differences in the genetic background of the mice used in the two studies explain the discrepant results is unclear. On the other hand, our discovery that GS and Cyp2e1 protein expression was relatively normal in the *Lyve1-Cre;Wls$^{f/f}$* newborn liver indicates that other cellular sources of Wnts (most probably CEVs) and not LSECs initiate metabolic zonation. Moreover, our results from in situ hybridization experiments and qRT-PCR analysis of isolated LSECs (CD117$^+$/CD31$^+$ and Lyve-1$^+$/CD31$^+$/CD117$^{LOW}$) support a model in which CEVs produce Wnt ligands and agonists (most likely Wnt2, Wnt9b and Rspo3) that induce metabolic Zones 2/3 in the perinatal liver, and LSECs secrete Wnt ligands (mainly Wnt2) that preserve metabolic and non-metabolic zonation in the adult liver. Overall, our results support and expand other published data (*Ding et al., 2010*; *Wang et al., 2015*; *Zhao et al., 2019*).

We also took advantage of *Cldn2-EGFP* mice to analyze simultaneously parenchymal and non-parenchymal cell behaviors in the CCl$_4$-acute injury mouse model. Our analysis revealed several interesting new findings in this model, including a small population of intact Cyp2e1$^+$/GFP$^+$ hepatocytes that remained located at the margins of the area of necrosis. Why some Cyp2e1$^+$ hepatocytes survive the injurious insult is not known, but one possibility is that the dose of CCl$_4$ is too low to access all Zone 2 hepatocytes. On the other hand, some results in our study indicated that the remaining Cyp2e1$^+$ hepatocytes and the surrounding Cyp2e1$^-$ hepatocytes repaired the damaged Zone 2 through proliferation: 1) those cells expressed the regulator of hepatic progenitor proliferation Tbx3 [(*Lüdtke et al., 2009*; 2) both Cyp2e1$^+$ hepatocytes and Cyp2e1$^-$ hepatocytes incorporated EdU in their DNA and retained this label for up to 5 days. This notion is favored by the results of recent studies showing that liver repair mainly occurs via proliferation of perivenous and periportal hepatocytes in different mouse models of liver injury (including the CCl$_4$ model described here) (*Chen et al., 2020*; *Matsumoto et al., 2020*; *Sun et al., 2020*). Interestingly, we found that in contrast to Cyp2e1$^+$/GFP$^+$ hepatocytes all hepatocytes in pericentral Zone 3 (GS$^+$/GFP$^+$) were destroyed upon CCl$_4$ acute exposure and in the next 1–2 days GS expression was induced de novo in a few GFP$^+$ hepatocytes located at the front of Zone 2. The number of GS$^+$/GFP$^+$ hepatocytes subsequently increased and those cells gradually adjoined the central vein endothelium as the

immune infiltrates resolved. These results demonstrate that Zone 3 is specified de novo at the front of an expanding Zone 2 during the repair process that follows CCl$_4$-acute liver injury. Similar observations were reported in a recent study in which Axin2$^+$ hepatocytes were genetically ablated and the authors showed complete destruction of Zone 3 and subsequent reappearance of GS$^+$ hepatocytes in areas juxtaposed to the CV endothelium (*Sun et al., 2020*). We hypothesize that Wnt ligands and Wnt agonists produced by intact LSECs are major players in the induction of a new Zone 3 in the CCl$_4$-injury model because the new GS$^+$ cells were always juxtaposed or near the liver sinusoidal endothelium and, conversely, those cells were physically separated from the central vein endothelium by macrophage infiltrates. Whether a specialized subset of LSECs produces the Wnt ligands and agonists that repair Zones 2 and 3 remains an open question although our results of immunofluorescence and magnetic bead purification experiments suggested that CD117$^+$/Lyve-1$^+$ LSECs are major sources of Wnt2, Wnt9b and Rspo3 during the repair process that follows CCl$_4$-induced injury.

In summary, our study complements and expands current knowledge on liver zonation and regeneration by showing that multiple Wnt endothelial sources help to orchestrate the spatiotemporal distribution of hepatocyte functions during liver maturation and to promote metabolic zone respecification during liver repair. The advantages we encountered using *Cldn2-GFP* mice advocate their use in similar investigations of parenchymal and non-parenchymal cell behavior in chronic liver damage or conditions that lead to pathologic claudin-2 expression.

# Materials and methods

## Key resources table

| Reagent type (species) or resource | Designation | Source/reference | Identifiers | Additional Information |
|---|---|---|---|---|
| Commercial assay or kit | Click-iT EdU Imaging Kit | Invitrogen | Invitrogen: C10338 | |
| Commercial assay or kit | PureLink RNA Mini Kit | Invitrogen | Invitrogen: 12183018A | |
| Commercial assay or kit | iScript cDNA Synthesis Kit | Biorad | Biorad: 170–8891 | |
| Commercial assay or kit | ALT Activity Assay | Sigma | Sigma: MAK052 | |
| Commercial assay or kit | AST Activity Assay | Sigma | Sigma: MAK055 | |
| Commercial assay or kit | ABC reagent | Vector Laboratories | Vector: PK-6100 | |
| Commercial assay or kit | DAB solution | Vector Laboratories | Vector: SK-4105 | |
| Commercial assay or kit | Pierce ECL Plus Western blotting Substrate | Pierce | Pierce: 32132 | |
| Commercial assay or kit | RNAscope 2.5 HD Duplex manual assay | Advanced cell diagnostics | Cat. #: 322436 | |
| Commercial assay or kit | DynabeadsTM FlowComp TM MouseCD4 kit | Invitrogen | Cat. #: 11461D | |
| Probe | RNAscope Probe Mm-Wnt2 | Advanced cell diagnostics | Cat #: 313601 | NM_023653.5, region 857–2086 |
| Probe | RNAscope Probe Mm-Rspo3-O2 | Advanced cell diagnostics | Cat. #: 483781 | NM_028351.3, region 717–2099 |
| Probe | RNAscope Probe Mm-Wnt9b | Advanced cell diagnostics | Cat.#: 405091 | NM_011719.4, region 706–1637 |
| Probe | RNAscope Probe Mm-Lyve1-C2 | Advanced cell diagnostics | Cat. #: 42451-C2 | NM_053247.4, region 2–952 |

*Continued on next page*

Continued

| Reagent type (species) or resource | Designation | Source/reference | Identifiers | Additional Information |
|---|---|---|---|---|
| Probe | RNAscope Probe Positive Control Probe | Advanced cell diagnostics | Cat. #: 320761 | Mm-Polr2a, NM_001291068.1, region 3212–4088 |
| Probe | RNAscope 2-Plex Negative Control | Advanced cell diagnostics | Cat. #: 320751 | DapB, CP015375.1, region 2252107–2252555 |
| Antibody | anti-mouse GFP (chicken polyclonal) | Abcam | Cat. #: ab13970; RRID:AB_300798 | IF, IHC (1:1000) |
| Antibody | anti-mouse GS (rabbit polyclonal) | Abcam | Cat. #: ab49873; RRID:AB_880241 | IF, IHC (1:5000) |
| Antibody | anti mouse Cyp2e1 (rabbot polyclonal) | Abcam | Cat. #: ab28146; RRID:2089985 | IF, IHC (1:500) |
| Antibody | anti-mouse E-cadherin (rat monoclonal) | Novex | Cat. #: 13–1900; RRID:AB_2533005 | IF (1:5000) |
| Antibody | anti-mouse HNF-4α (goat polyclonal | Santa Cruz Biotechnology | Cat. #: sc-6556; RRID:AB_2117025 | IF (1:50) |
| Antibody | anti-mouse Prox1 (rabbit polyclonal) | Proteintech | Cat. #: 11067–2-AP; RRID:AB_2268804 | IF (1:1000) |
| Antibody | anti-mouse Tbx3 (rabbit polyclonal) | Abcam | Cat. #: ab99302; RRID:AB_10861059 | IF (1:100) |
| Antibody | anti-mouse APC (rabbit polyclonal) | Abcam | Cat. #: ab52223; RRID:AB_867687 | IF (1:50) |
| Antibody | anti-mouse F4/80 (rat monoclonal) | Abcam | Cat. #: ab6640; RRID:AB_1140040 | IF (1:1000) |
| Antibody | anti-mouse PECAM-1 (rat monoclonal) | BD Pharmingen | Cat. #: 550274; RRID:AB_393571 | IF (1:100) MI (2 µg/25 µL) |
| Antibody | anti-mouse Lyve1 (goat polyclonal) | R and D Systems | Cat. #: BAF2125; RRID:AB_2138529 | IF (1:250) |
| Antibody | anti-mouse Claudin-2 (rabbit polyclonal) | Invitrogen | Cat. #: 51–6100; RRID:AB_2533911 | IHC (1:250) |
| Antibody | anti-mouse CD86 (rat monoclonal) | SouthernBiotech | Cat. #: 1735–01; RRID:AB_2795211 | IF (1:100) |
| Antibody | anti-mouse PCK1 (rabbit polyclonal) | Abcam | Cat. #: ab28455; RRID:AB_777191 | IHC (1:100) |
| Antibody | anti-mouse Beta-gal (chicken polyclonal) | Abcam | Cat. #: ab9361; RRID:AB_307210 | IF (1:2000) |
| Antibody | anti-mouse Lyve1 (rat monoclonal) | R and D systems | Cat. #: MAB215, RRID:AB_2138528 | MI (2 µg/25 µL) |
| Antibody | anti-mouse CD117 (rat monoclonal) | R and D Systems | Cat. #: MAB1356; RRID:AB_2131131 | IF (1:50) MI (2 µg/25 µL) |
| Antibody | Cy3 Anti-Rabbit IgG (H+L) (donkey polyclonal) | Jackson ImmunoResearch | Cat. #: 705-165-152; RRID:AB_2307443 | IF (1:250) |
| Antibody | Cy3 Anti-Goat IgG (H+L) (donkey polyclonal) | Jackson ImmunoResearch | Cat. #: 705-165-147; RRID:AB_2307351 | IF (1:250) |
| Antibody | Cy3 Anti-Rat IgG (H+L) (donkey polyclonal) | Jackson ImmunoResearch | Cat. #: 712-165-153; RRID:AB_2340667 | IF (1:250) |
| Antibody | Alexa Fluor 488 Anti-Chicken IgY (IgG) (H+L) (donkey polyclonal) | Jackson ImmunoResearch | Cat. #: 703-545-155; RRID:AB_2340375 | IF (1:250) |
| Antibody | Alexa Fluor 488 Anti-Rat IgG (H+L) (donkey polyclonal) | Jackson ImmunoResearch | Cat. #: 712-545-153; RRID:AB_2340684 | IF (1:250) |
| Antibody | Alexa Fluor 488 Anti-Goat IgG (H+L) (donkey polyclonal) | Jackson ImmunoResearch | Cat. #: 705-545-147; RRID:AB_2336933 | IF (1:250) |

*Continued*

| Reagent type (species) or resource | Designation | Source/reference | Identifiers | Additional Information |
|---|---|---|---|---|
| Antibody | Biotin-SP (long spacer) Anti-Rabbit IgG (H+L) (donkey polyclonal) | Jackson ImmunoResearch | Cat. #: 711-065-152; RRID:AB_2340593 | IHC (1:250) |
| Sequence based reagent | Glul_F | This paper | PCR primers | TGAACAAAGGCATCAAGCAAATG |
| Sequence-based reagent | Glul_R | This paper | PCR primers | CAGTCCAGGGTACGGGTCTT |
| Sequence-based reagent | Cyp2e1_F | This paper | PCR primers | CGTTGCCTTGCTTGTCTGGA |
| Sequence-based reagent | Cyp2e1_R | This paper | PCR primers | AAGAAAGGAATTGGGAAAGGTCC |
| Sequence- based reagent | Axin2_F | This paper | PCR primers | TGACTCTCCTTCCAGATCCCA |
| Sequence-based reagent | Axin2_R | This paper | PCR primers | TGCCCACACTAGGCTGACA |
| Sequence -based reagent | Cldn2_F | This paper | PCR primers | CAACTGGTGGGCTACATCCTA |
| Sequence-based reagent | Cldn2_R | This paper | PCR primers | CCCTTGGAAAAGCCAACCG |
| Sequence -based reagent | Tbx3_F | This paper | PCR primers | AGATCCGGTTATCCCTGGGAC |
| Sequence based reagent | Tbx3_R | This paper | PCR primers | CAGCAGCCCCCACTAACTG |
| Sequence-based reagent | Cdh1_F | This paper | PCR primers | CCAAGCACGTATCAGGGTCA |
| Sequence-based reagent | Cdh1_R | This paper | PCR primers | ACTGCTGGTCAGGATCGTTG |
| Sequence -based reagent | Prox1_F | This paper | PCR primers | AAGCGCAATGCAGGAAGGGCT |
| Sequence-based reagent | Prox1_R | This paper | PCR primers | ACCACTTGATGAGCTGCGAGG |
| Sequence-based reagent | Actb_F | This paper | PCR primers | AGATCAAGATCATTGCTCCTCCT |
| Sequence-based reagent | Actb_R | This paper | PCR primers | ACGCAGCTCAGTAACAGTCC |
| Sequence-based reagent | Apc_F | This paper | PCR primers | CTTGTGGCCCAGTTAAAATCTGA |
| Sequence-based reagent | Apc_R | This paper | PCR primers | CGCTTTTGAGGGTTGATTCCT |
| Sequence-based reagent | Wnt2_F | This paper | PCR primers | TCCGAAGTAGTCGGGAATCG |
| Sequence-based reagent | Wnt2_R | This paper | PCR primers | GCCCTGGTGATGGCAAATAC |
| Sequence-based reagent | Wnt9b_F | This paper | PCR primers | GGGCATCAAGGCTGTGAAGA |
| Sequence-based reagent | Wnt9b_R | This paper | PCR primers | AACAGCACAGGAGCCTGACA |
| Sequence-based reagent | Rspo3_F | This paper | PCR primers | ACACCTTGGAAAGTGCCTTGA |
| Sequence-based reagent | Rspo3_R | This paper | PCR primers | GCCTCACAGTGTACAATACTGACACA |
| Sequence-based reagent | Pecam1_F | This paper | PCR primers | AGCCTAGTGTGGAAGCCAAC |

*Continued on next page*

Continued

| Reagent type (species) or resource | Designation | Source/reference | Identifiers | Additional Information |
|---|---|---|---|---|
| Sequence-based reagent | Pecam1_R | This paper | PCR primers | GGAGCCTTCCGTTCTTAGGG |
| Sequence-based reagent | Kit_F | This paper | PCR primers | CAGGAGCAGAGCAAAGGTGT |
| Sequence-based reagent | Kit_R | This paper | PCR primers | GGGCCTGGATTTGCTCTTTG |
| Sequence-based reagent | Lyve1_F | This paper | PCR primers | GCCAACGCGGCCTGTAAGAT |
| Sequence-based reagent | Lyve1_R | This paper | PCR primers | CCCAGGTGTCGGATGAGTTG |
| Sequence-based reagent | Dll4_F | This paper | PCR primers | TGTGATTGCCACAGAGGTATAAGG |
| Sequence-based reagent | Dll4_R | This paper | PCR primers | GCAATGTAAACAATGCAGAAGGAA |
| Cell line (*Mus musculus*) | *Mus musculus* AML12 Cell line | ATCC | Cat. #: CRL-2254; RRID:CVCL_0140 | |
| Cell culture media | DMEM F12 | Gibco | Cat. #: 11320–033 | |
| Chemical compound, drug | Dexamethasone | Sigma-Aldrich | Cat. #: D4902 | (40 ng/ml) |
| Chemical compound, drug | 10 mg/ml insulin, 5.5 mg/ml transferrin, 5 ng/ml selenium | Gibco | Cat. #: 41400045 | (1X) |
| Chemical compound, drug | CHIR99021 | Sigma Aldrich | Cat. #: SML1046 | (3 µM) |
| Chemical compound, drug | CCl4 | Sigma Aldrich | Cat. #: 319961 | |
| Chemical compound, drug | Tamoxifen | Sigma Aldrich | Cat. #: T5648 | |
| Chemical compound, drug | Corn oil | Sigma Aldrich | Cat. #: C8267 | |
| Chemical compound, drug | Mayer's Hematoxylin | ScyTek Laboratories | Cat. #: HMM500 | |
| Chemical compound, drug | DAPI | Life Technologies | Cat. #: D1306 | (1 µg/mL) |
| Chemical compound, drug | TRIzol Reagent | Thermo Fisher | Cat. #: 15596026 | |
| Chemical compound, drug | cOmpleteTM, Mini EDTA-free protease inhibitor | Roche | Cat. #: 11836170001 | |
| Chemical compound, drug | DynabeadsTM Sheep Anti-Rat IgG | Invitrogen | Cat. #: 11035 | 25 µl/sample |
| Chemical compound, drug | Colagenase H from clostridium histolyticum | Millipore Sigma | Cat. #: 11074059001 | 0.5 mg/mL |
| Recombinant protein | Rspo3 | R and D Systems | Cat. #: 4120-RS-025 | (500 ng/ml) |
| Recombinant protein | Wnt2 | Abnova | Cat. #: H00007472-P01 | (500 ng/ml) |
| Recombinant protein | Wnt9b | R and D Systems | Cat. #: 3669-WN-025 | (500 ng/ml) |
| Strain *Mus musculus* | B6.129P2-Lyve1$^{tm1.1(EGFP/cre)Cys}$/J | The Jackson Laboratory | Cat. #: JAX:012601; RRID:IMSR_JAX:012601 | |

*Continued on next page*

*Continued*

| Reagent type (species) or resource | Designation | Source/reference | Identifiers | Additional Information |
|---|---|---|---|---|
| Strain *Mus musculus* | *Cdh5CreER*[T2] | Ralf H. Adams, Max Planck Institute for Molecular Biomedicine, Münster, Germany *Wang et al., 2010* | | |
| Strain *Mus musculus* | Tg[Cldn2-EGFP]OU78 Gsat/Mmucd | Mutant Mouse Regional Resource Center [MMRRC], University of California, Davis *Gong et al., 2003* | | |
| Strain *Mus musculus* | *129S-Wls*[tm1.1Lan/J] | The Jackson Laboratory | Cat. # JAX:012888; RRID:IMSR_JAX:012888 | |
| Strain *Mus musculus* | RoB6.129S4-Gt(ROSA) 26Sortm1Sor/Jsa | The Jackson Laboratory | Cat. # JAX:004077; RRID:IMSR_JAX:004077 | |
| Strain *Mus musculus* | *Sox9CreER*[T2] (*Tg(Sox9-cre/ERT2)*[1Msan/] | The Jackson Laboratory | Cat. # JAX:018829; RRID:IMSR_JAX:018829 | |
| Software | Adobe Photoshop CC | Adobe Systems | RRID:SCR_014199 | |
| Software | ImageJ | NIH https://imagej.net/ | RRID:SCR_003070 | |
| Software | GraphPad Prism 5.0 software | GraphPad Software; http://www.graphpad.com | RRID:SCR_002798 | |

## Mice

*Cldn2-EGFP* (Tg[Cldn2-EGFP]OU78Gsat/Mmucd; Mutant Mouse Regional Resource Center [MMRRC], University of California, Davis) were produced by the GENSAT Project (Rockefeller University, NY *Gong et al., 2003*). Ralf H. Adams (Max Planck Institute for Molecular Biomedicine, Münster, Germany) provided the *Cdh5CreERT2* mice (hereafter named as *Cdh5-CreERT2 Wang et al., 2010*). *Wls*[loxP/loxP] (*129S-Wls*[tm1.1Lan/J], stock 012888), *ROSA26-LacZ* (*B6.129S4-Gt(ROSA)26Sor*[tm1Sor/J], stock 003474), *ROSA26-EGFP* (*B6.129-Gt(ROSA)26Sor*[tm2Sho/J], stock 004077), *Sox9-CreERT2* (*Tg(Sox9-cre/ERT2)*[1Msan/J], stock 018829) and *Lyve1-Cre* (*B6.129P2-Lyve1*[tm1.1(EGFP/cre)Cys/J], stock 012601) mice were from The Jackson Laboratory (Bar Harbor, ME).

## Animal experiments

For adult *Wls* endothelial deletion, *Cdh5-CreERT2;Wls*[loxP/loxP] mice and *Cdh5-CreERT2* littermates (4–6 weeks old) received one intraperitoneal (i.p.) tamoxifen injection (250 mg/kg) for three consecutive days, and used for further experiments after a 2 week wash out period. For acute liver injury induction, an i.p. bolus of 1 µL/g body weight $CCl_4$ (Sigma, 319961, St. Louis, MO) diluted 4x in corn oil (Sigma, C8267, St. Louis, MO) was administered. For EdU labeling, mice first received an i.p. $CCl_4$ bolus (1 µL/g body weight) and 2 days later they were injected (i.p.) with EdU (50 µg/g body weight; Invitrogen A10044, Waltham, MA). For lineage tracing experiments using *Sox9-CreERT2; ROSA26-EGFP* mice, the animals first received an i.p. $CCl_4$ bolus (1 µL/g body weight) and for the next 3 days they received one daily intraperitoneal (i.p.) tamoxifen injection (250 mg/kg). Blood was collected from the retro-orbital plexus and coagulated at room temperature for 1 hr. Adult mice used in this study were age- and gender-matched littermates. The animal experiments are not randomized and the investigators were not blinded to allocation during experiments and outcome assessment.

## Morphometric analysis

Images were taken at 10x magnification for Zone 1 (Ecad[+]) and Zone 2/3 (Cyp2e1[+]/claudin-2/GFP[+]), and 20x for Zone 3 (GS[+]). Areas for quantification were selected using Photoshop CC 2015 and were measured using Image J software. Edu[+] hepatocytes were counted using ImageJ. Triplicate samples were used and at least three different areas were selected per sample for quantification.

## Immunofluorescence

Livers of embryos (embryonic [E] day E18.5), newborn (postnatal [P] day P2) or juvenile (P15) mice were harvested and fixed in 4% paraformaldehyde (PFA) at 4°C for 5 hr. Adult livers were isolated from mice perfused with 4% PFA, and post-fixed in 4% PFA at 4°C for 5 hr. Livers were then immersed in 30% sucrose overnight at 4°C, embedded in Tissue-Tek (Sakura, 25608–930, Torrance, CA) and cryosectioned (10 µm). Sections were incubated in blocking solution (Roche 11096176001, Mannheim, Germany) at room temperature for 30 min, then incubated with primary antibodies at room temperature overnight and with secondary antibodies for 2 hr at room temperature. Signals were developed with fluorescent-conjugated secondary antibodies (Alexa Fluor 488, Cy3 or Cy5; Jackson ImmunoResearch Laboratories) and nuclei were stained with DAPI (Cell Signaling Technology). Immunostaining images were obtained with a ZEISS Axioscop two fluorescence microscope and processed with Adobe Photoshop CC (Adobe Systems).

## Immunohistochemistry

Livers were isolated from newborn, P15 or adult mice and fixed with 4% PFA as indicated before. After embedding in paraffin, 7 µm sections were cut, deparaffinized in xylene, rehydrated in ethanol, and incubated with citrate antigen retrieval buffer in a 2100-Retriever (BioVendor Laboratory Medicine, Inc). After blocking, sections were incubated overnight at room temperature with primary antibodies followed by 2 hr incubation at room temperature with biotinylated secondary antibodies, and 30 min incubation at room temperature with ABC reagent (Vector Laboratories, PK-6100, Burlingame, CA). Sections were incubated with DAB solution (Vector Laboratories, SK-4105, Burlingame, CA) for visualizing the immunocomplexes, and counterstained with haematoxylin before mounting. Images were acquired with a ZEISS AXIOSCOP two fluorescence microscope and processed with Adobe Photoshop CC (Adobe Systems).

## Click-it edu incorporation assay

The Click-iT EdU reaction was performed as per the manufacturer's instructions (Click-iT EdU Imaging Kit, Invitrogen, C10338, Waltham, MA).

## Real-time PCR

RNA was isolated with PureLink RNA Mini Kit (Invitrogen, 12183018A). cDNA was synthesized using iScript cDNA Synthesis Kit (Biorad, 170–8891). The expression of mRNA for genes of interest was normalized to β-actin.

## Alanine aminotransferase (ALT) and Aspartate aminotransferase (AST) Assays

Whole blood was collected from the retroorbital plexus and coagulated at room temperature for 1 hr. The serum was separated by centrifugation at 3000 rpm at 4°C for 15 min. The serum levels of ALT and AST were measured using the ALT (Sigma, MAK052) and AST (Sigma, MAK055) kits according to the manufacturer's instructions.

## Cell culture

The AML12 mouse hepatocyte cell line was purchased from ATCC (ATCC CRL-2254) and cultured in DMEM F12 medium (Gibco, cat no. 11320–033) supplemented with 10% fetal bovine serum, 40 ng/ml dexamethasone (Sigma-Aldrich, catalog number D4902), 10 µg/ml insulin, 5.5 µg/ml transferrin and 5 ng/ml selenium (Gibco, catalog number 41400045). The cells were maintained in a humid, 5% $CO_2$ atmosphere. To stimulate β-catenin activity, we followed a protocol similar to that described in *Gerbal-Chaloin et al. (2014)*. Cells were plated in gelatin coated 6-well plates for 48 hr, fasted 24 hr in serum free medium, and incubated 24 hr with 3 µM CHIR99021 (GS3K inhibitor; Sigma-Aldrich, cat no. SML1046), 500 ng/ml Wnt2 (Abnova, cat no. H00007472-P01) or 500 ng/mL Wnt9b (R and D System, cat no. 3669-WN-025). For some experiments, the cells were incubated with Wnt2 or Wnt9b plus 500 ng/mL Rspo3 (R and D Systems, cat no. 4120-RS-025). RNA was isolated as described above and qRT-PCR was performed for *Axin2*, *Cyp2e1*, *Glul* and *Cldn2*. *Actb* expression was used for normalization.

### In situ hybridization

Livers were isolated from newborn (P2) or adult mice, cut in slices and fixed in 4% PFA at 4°C overnight. After embedding in paraffin, 7 µm sections were cut, deparaffinized in xylene, rehydrated in ethanol, and processed for RNA in situ hybridization using the RNAScope 2.5 HD Duplex Kit following the manufacturer's instructions (Advanced Cell Diagnostics). The following RNAscope probes were used: Wnt2 (Mm-Wnt2, Cat. 313601, NM_023653.5, region 857–2086), Wnt9b (Mm-Wnt9b, Cat. 405091, NM_011719.4, region 706–1637), Rspo3 (Mm-Rspo3-02, Cat. 483781, NM_028351.3, region 717–2099), Lyve1 (Mm-Lyve1-C2, Cat. 428451-C2, NM_053247.4, region 2–952), DapB (negative control, Cat. 320751, CP015375.1, region 2252107–2252555), Polr2a (positive control, Mm-Polr2a, Cat. 320761, NM_001291068.1, region 3212–4088).

### Hepatic endothelial cell (HEC) isolation with magnetic beads

Before HECs isolation, 2 µg of each rat anti-CD31(BD Pharmigen, cat. 550274), rat anti-CD117 (R and D Systems, cat. MAB1356) and rat anti-Lyve-1 (R and D systems, cat. MAB215) antibodies were incubated with 25 µl of magnetic beads (Dynabeads Sheep Anti-Rat IgG, Invitrogen, cat.11035) 1 hr at 4°C. A Dynamag-15 Magnet (Invitrogen, cat.12301D) was used to isolate the antibody conjugated-beads as per the manufacturer's recommendation. HECs were isolated from mice injected with $CCl_4$ or saline and sacrificed 3 days later. The liver was perfused through the inferior vena cava first with 20 mL perfusion medium I (1X PBS, 10 mM HEPES, 0.5 g/L KCl, 50 mM glucose, 0.2 mM EDTA, pH 7.4) at 37°C, followed by 50 mL perfusion medium II (1X PBS, 30 mM HEPES, 0.5 g/L KCl, 50 mM glucose, 1 mM $CaCl_2$, 0.5 mg/mL Collagenase H, pH 7.4) at 37°C. Upon dissecting the liver, the cells were dispersed in cold DMEM medium and passed through a 100 µm strainer. The nonparenchymal cells (NPCs) were separated from hepatocytes by low-speed centrifugation (50 x g, 5 min) at 4°C. The supernatant containing NPCs was collected and washed twice at 50 x g for 5 min at 4°C, and pelleted at 170 x g for 10 min. The cell pellets were resuspended in 1 mL red cell lysis buffer (155 mM $NH_4Cl$, 12 mM $NaHCO_3$, 0.1 mM EDTA) and incubated for 1 min at room temperature (RT), followed by addition of 10 mL isolation buffer (1X PBS $Ca^{2+}$ and $Mg^{2+}$ free, 0.1% BSA, 2 mM EDTA, pH 7.4). $1 \times 10^7$ NPCs were resuspended in 1 mL isolaton buffer and incubated with 25 µL of anti-CD31 Dynabeads for 30 min at 4°C, After the tube was placed in the magnet for 1 min, the supernatant was discarded, the magnetic beads were washed twice with isolation buffer and the $CD31^+$ cells were incubated with release buffer [Dynabeads FlowComp Mouse CD4, Invitrogen, 11461D] 10 min at RT, eluted and incubated with 25 µL of anti-CD117 Dynabeads for 30 min at 4°C. After Dynabead binding, the $CD117^+/CD31^+$ fraction was isolated as indicated above and lysed in TRIzol (1 mL) for RNA isolation. The unbound supernatant was incubated with anti-Lyve-1 Dynabeads and the $Lyve1^+/CD31^+/CD117^{low}$ cells were isolated as previously indicated.

### Statistics and quantitative analysis

The data represent the mean ± SEM and P value were calculated by two-tailed unpaired Student's t-test (*Figures 2E*, *3C, D*, *4F* and *6E*) or one-way ANOVA (*Figures 1D*, *4E*, *5E*, *6B* and *7C*) or two-way ANOVA (*Figures 3E, F* and *4B*) by the GraphPad Prism 8.0 software. NS, not significant (p>0.05), *p<0.05, **p<0.01, ***p<0.001. Each quantitative experiment was repeated at least three times. We considered biological replicates as those animals or tissues subjected to the same experimental test, and technical replicates as individual samples or tissues subjected to the same analysis.

### Study approval

All animal experiments were performed in accordance with protocols reviewed and approved by the Institutional Animal Care and Use Committee at Northwestern University. The animal welfare assurance number for this study is A3283-01.

## Acknowledgements

We thank Selina Begum, Ming-Yi Chiang and the Center of Comparative Medicine (Northwestern) for technical support, RH. Adams (Max Planck Institute for Molecular Biomedicine, Münster, Germany) for the *VE-cadherin-Cre^ERT2* mice, Karen M Ridge for assisting the magnetic bead cell isolation

experiments, and the GENSAT BAC Transgenic project (Rockefeller University, NY) for the *Cldn2-GFP* mice.

## Additional information

### Funding

| Funder | Grant reference number | Author |
|---|---|---|
| Feinberg School of Medicine | New Faculty Award 10040043-01 | Beatriz Sosa-Pineda |
| Consejo Nacional de Ciencia y Tecnología | Postdoctoral Fellowship (ASMR) | Angelica S Martínez-Ramírez |

Feinberg School of Medicine funded all the experiments associated with the study. CONACYT awarded a fellowship to Dr. Martinez-Ramirez The funders had no role in study design, data collection and interpretation, or the decision to submit the work for publication.

### Author contributions

Ruihua Ma, Conceptualization, Resources, Data curation, Formal analysis, Supervision, Validation, Investigation, Visualization, Methodology, Project administration; Angelica S Martínez-Ramírez, Conceptualization, Resources, Data curation, Formal analysis, Supervision, Validation, Investigation, Visualization, Methodology; Thomas L Borders, Resources, Formal analysis, Validation, Investigation, Visualization; Fanding Gao, Resources, Formal analysis, Validation, Investigation, Methodology; Beatriz Sosa-Pineda, Conceptualization, Resources, Data curation, Formal analysis, Supervision, Funding acquisition, Validation, Investigation, Visualization, Methodology, Project administration

### Author ORCIDs

Beatriz Sosa-Pineda (iD) https://orcid.org/0000-0001-6872-5444

### Ethics

Animal experimentation: This study was performed in strict accordance with the recommendations in the Guide for the Care and Use of Laboratory Animals of the National Institutes of Health. All of the animals were handled according to approved institutional animal care and use committee (IACUC) protocols (IS00003824, welfare assurance number A3283-01) of Northwestern University.

### Decision letter and Author response

Decision letter https://doi.org/10.7554/eLife.46206.sa1
Author response https://doi.org/10.7554/eLife.46206.sa2

## Additional files

### Supplementary files

• Transparent reporting form

### Data availability

All data generated or analysed during this study are included in the manuscript and supporting files. Source data files have been provided for Figures 1–7.

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
