## [Decision Letter]

**Acceptance summary:**

This paper includes a detailed description of establishment of lobular zonation in postnatal liver development and after liver injury. The role of Wntsignaling in this process and the importance of sinusoidal endothelial cells as a source of Wnts is specifically investigated. The results provide a reference point and tools for future studies of liver zonation.

**Decision letter after peer review:**

Thank you for submitting your article "Endothelial Wnt signaling establishes metabolic zonation during liver maturation and after acute hepatic injury" for consideration by *eLife*. Your article has been reviewed by three peer reviewers, including Holger Willenbring as the Reviewing Editor and Reviewer #1, and the evaluation has been overseen by Didier Stainier as the Senior Editor. The following individual involved in review of your submission has agreed to reveal their identity: Sabine Colnot (Reviewer #2).

The reviewers have discussed the reviews with one another and the Reviewing Editor has drafted this decision to help you prepare a revised submission.

Summary:

The manuscript by Ma et al. investigates the effect of endothelial cell-derived Wnts on zonation in liver maturation and regeneration. The authors present an analysis of how metabolic zonation in the mouse liver changes from the perinatal period to adulthood, using markers of Zones 1-3, including new *Cldn2-EGFP* mice. They show that glutamine synthetase and Cyp2e1 zonation is fixed early, whereas specific zonation of claudin-2, E-cadherin and PEPCK occurs later. The authors further show that deleting Wntless (Wls) in sinusoidal endothelial cells (LSECs) using *Lyve1-Cre* reduces Zones 2 and 3 and increases Zone 1 marker expression. They confirm these findings in adult mice using *VE-cadherin^creER^*, which deletes Wls in all endothelial cells. The authors also use *VE-cadherin^creER^;Wls* mice to show that Wnts secreted by endothelial cells are needed to re-establish Zone 3 marker expression after CCl_4_ injury. The detailed description of the development and maintenance of metabolic zonation is convincing and interesting. The authors show for the first time that Wnt secretion by LSECs rather than central vein endothelial cells (CEVs) is responsible for liver zonation. Which Wnts are involved and how they shape the different expression patterns of zonated genes is not investigated. It is possible that zonated genes differ in their responsiveness to Wnts or that LSECs differ in which and how much Wnt they secrete. Addressing these questions would provide valuable mechanistic insight. The manuscript would also benefit from more detailed characterization of the new *Cldn2-EGFP* mouse. The Discussion should be revised to be more focused and put the new findings in perspective with the existing literature. Along these lines, some interesting but descriptive observations should be interpreted more carefully or removed.

If certain experimental revisions cannot be carried out, please provide clear arguments explaining why these experiments are not possible in the Rebuttal letter.

Essential revisions:

1) Quantification of Wnts and R-spondin3 in LSECs in all 3 zones, CEVs and portal vein endothelial cells in the perinatal and adult normal liver and in the CCl_4_-injured liver, with and without Wls inactivation, e.g., using RNA in situ hybridization.

2) Analysis of whether increasing Wnt levels affects the expression patterns of glutamine synthetase, Cyp2e1 and claudin-2, e.g., in vitro.

3) Detailed characterization of *Cldn2-EGFP* mice: Overlap of claudin-2 and EGFP expression should be analyzed at different time points, e.g., using RNA in situ hybridization. Whether claudin-2 and Cyp2e1 are invariably co-expressed or whether some Cyp2e1^+^ cells lack claudin-2 expression should be clarified. Co-staining of claudin-2 with a biliary marker should be added.

4) *Lyve1-Cre;Wls* mice and also *VE-cadherin^creER^;Wls* mice are shown to be resistant to CCl_4_ liver damage. It is hypothesized that reduction of Zones 2 and 3 and thus Cyp2e1 expression leads to insufficient CCl_4_ activation in these mice. This could be tested by expressing Cyp2e1 in these zones, e.g., using AAV8 vectors, which preferentially transduce these zones.

5) Interesting but descriptive observations that should be interpreted more carefully or removed:

– The part of the manuscript about the source of new hepatocytes after CCl_4_ injury states that some Zone 2 cells are spared despite expressing the CCl_4_ activator Cyp2e1 and that these cells are the main drivers of regeneration, although periportal hepatocytes also proliferate. Because these results were obtained using EdU labeling, it is possible that these cells underwent polyploidization instead of division, which is not addressed.

– Similarly, it is concluded that stem cells or progenitor cells do not contribute to regeneration after CCl_4_ injury but no data is shown.

6) In the Discussion, the new results should be put into perspective with existing literature, including recent papers like PMID 30762896. For example, it should be discussed why *VE-cadherin^creER^;Wls* mice acutely died after injection of CCl_4_ followed by tamoxifen in that paper but not in the authors' study.

7) The title should be revised to highlight the manuscript's novel findings.

[Editors' note: further revisions were suggested prior to acceptance, as described below.]

Thank you for submitting your article "Metabolic and non-metabolic liver zonation is established non-synchronously and requires sinusoidal Wnts" for consideration by *eLife*. Your article has been reviewed by three peer reviewers, including Holger Willenbring as the Reviewing Editor and Reviewer #1, and the evaluation has been overseen by Didier Stainier as the Senior Editor. The following individual involved in review of your submission has agreed to reveal their identity: Sabine Colnot (Reviewer #2).

The reviewers have discussed the reviews with one another and the Reviewing Editor has drafted this decision to help you prepare a revised submission.

Summary:

Ma et al. report Wnt-dependent establishment of liver zonation in postnatal development and after injury, including essential contribution of LSECs to Wnt secretion. The revised manuscript is improved, but two key points remain to be addressed to support the conclusions and novelty.

Essential revisions:

1) Quantification of Wnt2, Wnt9b and Rspo3 expression in endothelial cells in normal and injured liver is still missing. The reviewers agree with the authors that analyzing these genes in Wls-deleted mice will probably not be informative; however, these genes should be analyzed and quantified in all three zones of the liver lobule in untreated and CCl_4_-treated wildtype mice. A minimum of 3 mice per group should be analyzed.

2) in vitro testing of Wnt2 (not Wnt3), with and without Rspo3. This additional experiment will better integrate the in vitro data with the in vivo data and potentially clarify or substantiate the findings related to how LSECs contribute to liver zonation, e.g., whether there is Wnt/Rspo redundancy. The in vitro LiCl data could be removed because this compound lacks specificity.

---

## [Author Response]

Essential revisions:1) Quantification of Wnts and R-spondin3 in LSECs in all 3 zones, CEVs and portal vein endothelial cells in the perinatal and adult normal liver and in the CCl_4_-injured liver, with and without Wls inactivation, e.g., using RNA in situ hybridization.

As requested by the reviewers, we performed in situ hybridization experiments in perinatal (P2) and adult (P30) wildtype livers to investigate the expression of Rspo3, Wnt2 and Wnt9b in CEVs and LSECs. We did not quantify these results because we did not have sufficient material for a quantitative analysis (the reason being that we had to purchase a complete RNAscope system and the included kit only has sufficient reagents to stain 20 slides). Also, although initially we considered using our regular in situ hybridization protocol with digoxigenin-labelled probes we did not succeed obtaining plasmids for the 3 probes from other collaborators (and we did not attempt to prepare the vectors ourselves due to time limitations). Moreover, we obtained an in situ hybridization protocol suitable for adult mouse livers from the reviewer that we tested using an Alb probe and worked very well. However, when we used a DIG-labeled Rspo3 probe validated in embryonic tissues we did not obtain a signal using sections of the same liver. Notwithstanding, our double RNAscope in situ results using a Lyve1 probe combined with Wnt2, Rspo3 or Wnt9b probes were successful since, as reported recently by the R. Nusse’s group (Zhao et al., 2019) we demonstrated expression of Wnt2, Rspo3 and Wnt9b in the central vein endothelium (new Figure 3A). More important, we conclusively demonstrate expression of those transcripts in perivenous LSECs in P2 and P30 livers (new Figure 3A).

Finally, we did not perform the suggested in situ experiments in wildtype and Wls-deleted CCl_4_ livers for 2 reasons: 1) since Zhao et al., 2019, already reported the expression of various Wnts in PECAM^+^ endothelial cells in the CCl_4_-injured liver, the results of the proposed experiment would not be completely novel; 2) we did not perform an in situ analysis in the Wls-depleted liver since this alteration mainly blocks the secretion of Wnts and probably does not affect the expression of these ligands or Rspo3 (unless there is some sort of feedback effect, something that is beyond the scope of our study).

2) Analysis of whether increasing Wnt levels affects the expression patterns of glutamine synthetase, Cyp2e1 and claudin-2, e.g., in vitro.

In new Figure 3 (“Lack of Wnt ligand secretion from LSECs impairs adult zonation maintenance”) we added a new result (Figure 3F) showing how Wnt/β-catenin stimulation affects *Axin2*, *Cyp2e1*, *Glul* and *Cldn2* transcript expression in the Wnt-responsive mouse hepatocyte cell line AML12. These new results show that culturing AML12 cells with known Wnt pathway stimulators (i.e., LiCl, CHI99021 or Wnt3a) significantly increases expression of the canonical target *Axin2*. Those treatments also had variable effect on *Cyp2e1* expression and only minimally affected *Glul* and *Cldn2* expression. We discussed these new results in the Discussion section in the amended manuscript.

3) Detailed characterization of Cldn2-EGFP mice: Overlap of claudin-2 and EGFP expression should be analyzed at different time points, e.g., using RNA in situ hybridization. Whether claudin-2 and Cyp2e1 are invariably co-expressed or whether some Cyp2e1^+^ cells lack claudin-2 expression should be clarified. Co-staining of claudin-2 with a biliary marker should be added.

The new Figure 1 and its associated supplementary figures, Figure 1—figure supplements 1-3 describe in detail the expression of claudin-2/GFP in the liver of *Cldn2-EGFP* mice at various stages: E18.5, P2, P15, P30 and 6 months of age. In liver sections, we used double-immunofluorescence to show the extent of overlap of claudin-2/GFP with *Cyp2e1* and these results are quantified in new Figure 1D. We did not perform in situ experiments using *Cldn2* and *Cyp2e1* probes due to the limited reagents in each RNAscope kit. Moreover, for these experiments we have to isolate new livers from *Cldn2-EGFP* mice of different ages since all the specimens that we used for immunostaining analysis were prepared using conditions that are not suitable for in situ analysis. Also, it was not possible to stain Claudin2-EGFP liver sections with chicken anti-GFP and rabbit anti-claudin-2 antibodies since these reagents use conditions that are incompatible (i.e., paraffin vs. frozen sections). However, we believe that the results in new Figure 1A convincingly show the identical distribution of GFP and claudin-2 proteins in newborn and adult livers. Moreover, new Figure 1—figure supplement 1 demonstrates identical expression of GFP and claudin-2 in the intrahepatic bile ducts and gall bladder of mice.

4) Lyve1-Cre;Wls mice and also VE-cadherin^creER^;Wls mice are shown to be resistant to CCl_4_ liver damage. It is hypothesized that reduction of Zones 2 and 3 and thus Cyp2e1 expression leads to insufficient CCl_4_ activation in these mice. This could be tested by expressing Cyp2e1 in these zones, e.g., using AAV8 vectors, which preferentially transduce these zones.

We agree that we need more experimental evidence in support that *Lyve1-Cre;Wls* mice and *VE-cadherin^creER^;Wls* mice are resistant to CCl_4_-induced injury because their liver has low *Cyp2e1* expression. We also agree that performing the suggested *AAV8-Cyp2e1* experiments should be an appropriate way to test this hypothesis. However, the preparation and isolation of the AAV8 constructs will be time consuming and currently only 2 people in my laboratory are helping the paper’s resubmission since the postdoc involved in this project left the lab in May. Therefore, as I decided to focus on the most relevant aspects of the story the previous conclusion was removed in the new Discussion section and is briefly alluded to it at the end of the new Results subsection “These unexpected results suggested that although some *Cyp2e1*^+^ hepatocytes remain in *Lyve1-cre;Wls^f/f^* livers, these cells are protected or refractory to CCl_4_-induced toxicity”.

5) Interesting but descriptive observations that should be interpreted more carefully or removed:– The part of the manuscript about the source of new hepatocytes after CCl_4_ injury states that some Zone 2 cells are spared despite expressing the CCl_4_ activator Cyp2e1 and that these cells are the main drivers of regeneration, although periportal hepatocytes also proliferate. Because these results were obtained using EdU labeling, it is possible that these cells underwent polyploidization instead of division, which is not addressed.– Similarly, it is concluded that stem cells or progenitor cells do not contribute to regeneration after CCl_4_ injury but no data is shown.

We agree that some of our previous interpretations were probably partially incorrect and therefore generated some confusion. Therefore, based on the most recent publication of the Nusse’s group (Zhao et al., 2019) and our new data on Tbx3 expression in the CCl_4_-injury liver, we modified our initial model of zonation recovery and propose that Zone 2 is restored via proliferation of (GFP^+^/*Cyp2e1*^+^) and (GFP^-^/*Cyp2e1*^-^) hepatocytes located around the damaged area, and Zone 3 is induced de novo at the margins of expanding Zone 2 (new Discussion section). I should also indicate that we performed lineage-tracing experiments using *Sox9^creER^;ROSA-EGFP* mice to investigate if *Sox9^+^* hepatocytes contribute to repopulate the damaged perivenous areas in the CCl_4_-acute injury model and our new results ruled out this possibility (new Figure 4—figure supplement 2).

6) In the Discussion, the new results should be put into perspective with existing literature, including recent papers like PMID 30762896. For example, it should be discussed why VE-cadherin^creER^;Wls mice acutely died after injection of CCl_4_ followed by tamoxifen in that paper but not in the authors' study.

We agree with the reviewers that is important to compare our findings with the existing literature. In this revision, we cite other related publications throughout the Results section to compare our new findings with published data. Also, the new Discussion section discusses both, similarities and differences between the recently published study from the Nusse’s group (PMID 30762896) and ours.

I should mention that the Zhao et al. study showed that Wls deletion using *VE-cadherin^creER^* reduces *Axin2* expression and EdU incorporation in hepatocytes mouse livers acutely damaged with CCl_4_. In contrast, our findings in new Figure 6 provides a detailed description of how endothelial Wls deletion affects zonation pattern recovery and Zone 3 restoration after acute CCl_4_ administration. Thus, our study significantly expands the observations in the Zhao et al. paper. On the other hand, the only explanation I have for the different survival of *VE-cadherin^creER^;Wls^f/f^*mice injected with CCl_4_ is that our respective studies used mice of different genetic background (mixed NMRI vs. mixed C57/BL6). The comparison of our results and those in the Zhao et al. study using *VE-cadherin^creER^;Wls^f/f^*mice injected with CCl_4_ is included in new Discussion section.

7) The title should be revised to highlight the manuscript's novel findings.

We thank the reviewers for their reasonable suggestion. The previous title “Endothelial Wnt signaling establishes metabolic zonation during liver maturation and after acute hepatic injury” has been changed to “Metabolic and non-metabolic liver zonation is established nonsynchronously and requires sinusoidal Wnts” to better highlight our novel findings.

[Editors' note: further revisions were suggested prior to acceptance, as described below.]Essential revisions:1) Quantification of Wnt2, Wnt9b and Rspo3 expression in endothelial cells in normal and injured liver is still missing. The reviewers agree with the authors that analyzing these genes in Wls-deleted mice will probably not be informative; however, these genes should be analyzed and quantified in all three zones of the liver lobule in untreated and CCl_4_-treated wildtype mice. A minimum of 3 mice per group should be analyzed.2) in vitro testing of Wnt2 (not Wnt3), with and without Rspo3. This additional experiment will better integrate the in vitro data with the in vivo data and potentially clarify or substantiate the findings related to how LSECs contribute to liver zonation, e.g., whether there is Wnt/Rspo redundancy. The in vitro LiCl data could be removed because this compound lacks specificity.

I would like to thank again the reviewers of our paper for their valuable criticism and insightful suggestions. Their latest revision mentioned the need to fill some important gaps and asked that we perform 2 additional experiments: 1) Analyzing and quantifying (in triplicate) the expression of Wnt2, Wnt9b and Rspo3 in endothelial sinusoidal cells located in Zones 1-3, in control and CCl_4_ injured livers; 2) Repeating the in vitro experiments in AML12 cells using Wnt2 instead of Wnt3 and with and without Rspo3.

For the first experiment, the reviewers initially suggested to perform quantitative in situ hybridization experiments using RNAscope but I mentioned that those experiments would be quite expensive. Therefore, they suggested instead FACS isolation of CD117+ liver endothelial cells as this marker is zonated in liver sinusoidal cells or LSECs. We followed their advice and used magnetic beads to separate CD31+/CD117^HIGH^ (‘pericentral’) endothelial cells from Lyve1^+^/CD31^+^/CD117^LOW^ (‘mid-zonal/periportal’) LSECs. The experiments were repeated in 3 individual control livers and 3 individual CCl_4_-injured livers. Similar to the recent Halpern et al. paper (Halpern et al., 2018), the isolated CD117^HIGH^ and CD117^LOW^ endothelial cells were LSECs as they expressed Lyve-1. Also, we detected differences in Wnt2, Wnt9b and Rspo3 transcript expression between CD31+/Lyve1+/CD117^HIGH^ cells and Lyve1+/CD31+/CD117^LOW^ cells isolated from control livers that matched the former published scRNAseq results. Perhaps more interesting, we found that Wnt2, Wnt9b and Rspo3 transcripts were upregulated in both CD31+/Lyve1+/CD117^HIGH^ cells and Lyve1+/CD31+/CD117^LOW^ cells from CCl_4_-injured livers. Our interpretation of these quantitative results is that Wnt signaling is upregulated along the hepatic sinusoids upon acute liver injury. Also, to complement this analysis we performed immunofluorescence experiments and demonstrated that CD117 is expressed in pericentral/perivenous liver sinusoidal cells but not in central vein endothelial cells in both, control livers and CCl_4_-injured livers. The new results are now part of new Figure 6C-E and the conclusions from these experiments are incorporated in the new Discussion section.

In regards to the second point, we performed the suggested experiments in AML12 cells and included the new results in panel E of new Figure 3. These results demonstrate stimulation of *Axin2* and *Cyp2e1* expression with Wnt2 and Wnt9b and a synergistic effect of Rspo3. In contrast, these experiments did not show an obvious effect of the Wnt ligands/agonist on *Glul* or *Cldn2* expression. Several possible explanations for these discrepant results are considered in the new Discussion section of our manuscript, including: differences in the threshold and/or combination of Wnt ligands needed to stimulate the expression of those genes; the possibility that other pathways cooperate with Wnt/β-catenin to stimulate *Cldn2* expression (especially since we found that a GSK3 inhibitor increases *Cldn2* expression in AML12 cells), and the fact that AML12 is an immortalized hepatocyte cell line. These new results are also discussed in the amended version of the manuscript.

In sum, all the new results are shown in Figure 3E and Figure 6C-E, and the text changes are indicated in blue in the revised article file. I hope that our new results answer the remaining concerns of the reviewers and they find our latest manuscript suitable for publication in *eLife*. I am very grateful for their valuable advice, their suggestions on how to investigate the expression of Wnts in endothelial sinusoidal cells, and for agreeing to disclose their names.